# YuE: Scaling Open Foundation Models for Long-Form Music Generation

✴ **Multimodal Art Projection**[*], **Hong Kong University of Science and Technology**[*]

## Abstract

We tackle the task of long-form music generation, particularly the challenging **lyrics-to-song** problem, by introducing **YuE** (乐), a family of open-source music generation foundation models. Specifically, YuE scales to trillions of tokens and generates up to five minutes of music while maintaining lyrical alignment, coherent musical structure, and engaging vocal melodies with appropriate accompaniment. It achieves this through **track-decoupled next-token prediction** to overcome dense mixture signals, and **structural progressive conditioning** for long-context lyrical alignment. In addition, we redesign the **in-context learning** technique for music generation, enabling bidirectional content creation, style cloning, and improving musicality. Through extensive evaluation, we demonstrate that YuE matches or even surpasses some of the proprietary systems in musicality and vocal agility (as of 2025-01). We strongly encourage readers to **listen to our demo**[1].

## 1 Introduction

Neural music generation is transforming artistic creativity and holds significant commercial and cultural potential (Ma et al., 2024). Among music generation tasks, lyrics-to-song, creating full songs with vocals and accompaniment from lyrics, remains highly challenging. By the release time of our system, no open-source system can reliably achieve this at scale, restricting innovation and accessibility despite proprietary solutions like Suno and Udio[2] demonstrating promising results.

Key challenges of lyrics-to-song generation include: 1) *Long-range dependencies*: Music exhibits complex temporal structures spanning several minutes, making it difficult for models to maintain coherence over extended durations. 2) *Signal complexity*: Unlike speech or environmental sounds, music is inherently polyphonic, requiring precise coordination between multiple instrumental and vocal components. 3) *Linguistic distortion*: Singing alters phonemes, durations, and prosody in ways that differ significantly from spoken language, complicating the alignment between lyrics and melody. 4) *Data scarcity*: The lack of large-scale, high-quality paired datasets of lyrics, vocals, and accompaniment limits model training and generalization capabilities.

**Related Work.** Existing models for music generation and singing voice synthesis (SVS) are often limited in several key aspects. For instance, models such as Jukebox (Dhariwal et al., 2020) and MusicLM (Agostinelli et al., 2023) typically generate instrumental music with short durations (around 30 seconds) and lack coherent lyrical semantics when attempting to incorporate vocals. While some SVS systems (Chen et al., 2020; Liu et al., 2022) have been successful in generating vocals, they often rely on explicit melodic guidance and do not seamlessly integrate vocals with instrumental accompaniments, especially over long-form durations. Moreover, academic models such as MelodyLM (Li et al., 2024) and SongCreator (Lei et al., 2024) struggle to maintain high-quality, long-duration synthesis beyond 30 seconds, often suffering from artifacts or limited controllability. Industry-developed systems, such as Tiangong Music and Suno, offer better performance but remain proprietary, with limited technical transparency.

---

[*]Co-first affiliation. These institutions contributed equally.
[1]https://map-yue.github.io/
[2]https://suno.com/, https://www.udio.com/

In this paper, we introduce **YuE**, the first[3] open-source foundation model family for high-quality, full-song lyrics-to-song generation. Built upon LLaMA2 architechture (Touvron et al., 2023b; Zhang et al., 2024) and trained on trillions of tokens, YuE generates coherent, lyrically-aligned, and musically interesting songs up to five minutes in length, matching or surpassing proprietary systems (Figure 3). Our main contributions include:

1) **Track-Decoupled Next-Token Prediction (NTP)**: A dual-token strategy that separately models different audio tracks (vocals, accompaniment) at the frame level, resilient to challenging low vocal-to-accompaniment ratio scenarios like metal (Section 2.2).

2) **Structural Progressive Conditioning**: A progressive conditioning strategy for long-form music generation, enabling song-level lyrics following and structure control (Section 2.3).

3) **Redesigned In-Context Learning (ICL) for Music**: A novel ICL framework enabling advanced style transfer, voice cloning, and bidirectional content creation (Section 2.4).

## 2 YuE

### 2.1 Overview

YuE is an autoregressive (AR) language model (LM)-based framework tailored for lyrics-to-song generation. As depicted in Figure 1, YuE comprises four main components: an audio tokenizer (with a lightweight upsampler), a text tokenizer, and two LMs. The audio tokenizer converts waveforms into discrete tokens using a semantic-acoustic fused approach. The Stage-1 LM (Figure 2) is track-decoupled, trained on text tokens and semantic-rich bottom-level audio tokens (codebook-0 from Residual VQ-VAE), modeling lyrics-to-song generation as an AR NTP task. In Stage-2, a smaller LM predicts residual tokens from codebook-0 tokens to reconstruct audio. Both LMs follow the widely-adopted LLaMA2 architecture (Touvron et al., 2023a; Team, 2024). Next, we will introduce the key techniques for training the Stage-1 LM.

### 2.2 Track-Decoupled Next-Token Prediction

**Challenges of Standard NTP.** Popular LM-based approaches for modeling long residual vector quantization (RVQ) sequences typically adopt a multi-stage design (Wang et al., 2023a; Agostinelli et al., 2023; Borsos et al., 2023), where the first stage commonly uses a single codebook-0 token to represent each audio frame.[4] Let $\mathbf{x}_{1:T} = (x_1, x_2, \dots, x_T)$ represent a sequence of audio tokens, where each $x_t$ corresponds to one frame. In a standard NTP framework, we factorize the joint probability of $\mathbf{x}_{1:T}$ as:

$$P(\mathbf{x}_{1:T}) = \prod_{t=1}^{T} P(x_t \mid x_{<t}; \theta), \qquad (1)$$

where $\theta$ is the model parameter. During inference (generation), the model predicts the next token $\hat{x}_t$ which maximizes the conditional distribution:

$$\hat{x}_t = \arg\max_{x_t} P(x_t \mid x_{<t}; \theta). \qquad (2)$$

Figure 1: Overview of YuE: two-stage lyrics-to-song generation. Stage-1: music LM. Stage-2: residual modeling. Blue: vocal tokens. Orange: accompaniment tokens. Grey: residual tokens.

---

[3]As of its release on Jan. 28, 2025, YuE familiy is the **first** publicly available, open-source lyrics-to-song model capable of full-song generation with quality on par with commercial systems.

[4]We acknowledge single-stage methods such as MusicGen, which utilize delay or parallel decoding patterns to reduce sequence length. However, we observed that the parallel decoding pattern fails to converge on our dataset, while the delay pattern results in longer sequences compared to multi-stage approaches.

This approach works well for tokens $\mathbf{x}_{1:T}$ representing purely vocal (text-to-speech, TTS) or instrumental (text-to-music, TTM) signals but struggles when encoding both vocals and accompaniment simultaneously due to differing dynamics, as in lyrics-to-song tasks combining TTS and TTM. We observed that RVQ compression causes significant linguistic information loss in genres with high acoustic complexity (*e.g.*, metal). More discussions in Appendix E.

**Track-Decoupled Next-Token Prediction (Dual-NTP).** The above challenge shows that the issue arises from forcing a single token $x_t$ to represent two distinct signals: vocal and music. Accompaniment can overshadow the vocal track, degrading lyric intelligibility. To overcome these limitations, we propose to explicitly use a source separation prior, splitting each time step into two tokens: one for **vocal** and one for **accompaniment** (see dotted token pairs in Figure 2).

In the proposed method, each time step $t$ outputs two tokens: $v_t$ (vocal token) and $a_t$ (accompaniment token). The model's sequence of tokens thus becomes:

$$\Big( \underbrace{v_1}_{\text{vocal}}, \underbrace{a_1}_{\text{accomp.}}, \underbrace{v_2}_{\text{vocal}}, \underbrace{a_2}_{\text{accomp.}}, \ldots, \underbrace{v_T}_{\text{vocal}}, \underbrace{a_T}_{\text{accomp.}} \Big). \tag{3}$$

To formally define this, let $\mathbf{v}_{1:T} = (v_1, v_2, \ldots, v_T)$ and $\mathbf{a}_{1:T} = (a_1, a_2, \ldots, a_T)$. We factorize their joint probability as:

$$P\big(\mathbf{v}_{1:T}, \mathbf{a}_{1:T}\big) = \prod_{t=1}^{T} P\Big(v_t, a_t \,\Big|\, v_{<t}, a_{<t}; \theta\Big). \tag{4}$$

Although this probability is written in joint form, it can be decomposed as:

$$P\Big(v_t, a_t \,\Big|\, v_{<t}, a_{<t}; \theta\Big) = P\Big(v_t \,\Big|\, v_{<t}, a_{<t}; \theta\Big) \times P\Big(a_t \,\Big|\, v_{\leq t}, a_{<t}; \theta\Big), \tag{5}$$

making it straightforward to implement in standard AR decoding frameworks.

**Discussion.** Prior work on dual-track modeling explores a range of designs (Lei et al., 2024; Li et al., 2024), but either requires substantial modifications to the base LM or treats the two tracks in a sequential pipeline, incurring latency and error propagation. Our approach jointly models both tracks with minimal architectural changes and no sequential dependency, yielding these advantages.

1) **Scalability:** By preserving the existing LM architecture, we leverage well-established pre-training infrastructures and enable straightforward scalability.

2) **Convergence:** Empirically, Dual-NTP converges to lower training loss compared to standard NTP (see Section 5). Notably, it demonstrates robust lyric adherence even within challenging minority genres (*e.g.*, metal music)[5].

3) **Joint Modeling of Tracks:** Our approach jointly contextualizes both tracks in a single forward pass, avoiding track synchronization issues, and allowing coherent and natural musical planning.

## 2.3 STRUCTURAL PROGRESSIVE CONDITIONING

**Challenges of Full-song Generation.** While typical TTS and TTM systems operate on less than 30 seconds of context (Liu et al., 2023; 2024b; Wang et al., 2023a; Borsos et al., 2023; Copet et al., 2023), full-song modeling requires handling minutes-long contexts. We find that extending the LM context to full-song modeling is non-trivial. The commonly used prefix text conditioning degrades as the length of audio tokens increases. Empirically, this degradation begins around 3K tokens and leads to complete failure beyond 6K tokens. Our mitigation attempts, such as increasing the RoPE base (from 10K to 100K) or curriculum learning with gradually increasing audio lengths, have been ineffective. See ablation in Section 5 for more details.

**Structural Progressive Conditioning (SPC).** To address this, we propose an elegant solution for Stage-1 LM that leverages the inherent structural priors of music. Songs are typically composed of distinct segments, such as intro, verse, chorus, bridge, and outro (Nieto et al., 2020; Bruderer et al., 2009; Lerdahl & Jackendoff, 1996). We use all-in-one (Kim & Nam, 2023) to automatically segment

---

[5]We encourage the readers to listen to our demo.

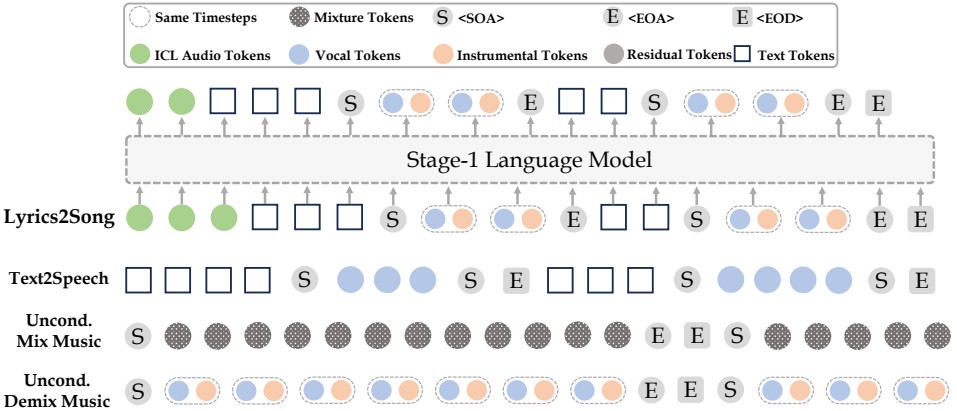

Figure 2: The Stage-1 Framework of YuE. Dotted lines: Dual-NTP (Section 2.2). Text interleave: SPC (Section 2.3). Green tokens: ICL (Section 2.4).

songs into musical sections, with most of the sections shorter than 30 seconds. Within each structure section, text conditions (*i.e.* lyrics and structure tags) and audio are paired together. From a full song perspective, structured text and audio tokens are **interleaved** (see lyrics2song token arrangement in Figure 2). In other words, each document in SPC begins with an instruction, tags, and raw lyrics, followed by a series of interleaved lyrics-audio segments.

## 2.4 MUSIC IN-CONTEXT LEARNING

**Deficiencies of Speech ICL.** Previous work in TTS (Wang et al., 2023a; Du et al., 2024) often defines speech ICL via a continuation-based approach. The sequence is constructed as:

$$\underbrace{T_{\text{ref}}}_{\text{reference text}} \circ \underbrace{T_{\text{input}}}_{\text{input text}} \circ \underbrace{A_{\text{ref}}}_{\text{reference audio}} \circ \underbrace{A_{\text{gen}}}_{\text{generated audio}}$$

While this framework can be suitable for speech-based tasks, there are three major issues when directly applying it to music:

1) **Necessity of reference text.** Requiring a text transcript for the reference audio can be redundant in a musical context, and lyrics may be unavailable or challenging to obtain.

2) **Unidirectional assumption.** Continuation is unidirectional and restricts the task generalization in scenarios requiring bidirectional creativity, *e.g.*, writing an entire piece from a short motif.

3) **Entanglement.** Continuation imposes strong constraints on the style and content of the generated audio. Given that music often features structural repetition, the model may simply replicate the reference melody or even entire segments, raising copyright concerns.

**Re-designing ICL for Music.** The aforementioned issues necessitate a novel approach to ICL for music. We propose a revised formulation of music ICL by extending the ICL format from SPC data. We randomly sample a 30s segment from the reference track and prepend its tokens to SPC data:

$$\mathcal{D}_{\text{icl}} = A_{\text{ref}} \circ \mathcal{D}_{\text{spc}}.$$

We find that this form of ICL can be effectively activated with minimal computational overhead (∼2% of the total pre-training cost). However, ICL constitutes a strong conditioning signal and can be considered as "easy" data. Our preliminary experiments reveal that incorporating ICL data too early encourages **shortcut learning** (Geirhos et al., 2020), where the model tends to directly copy the reference audio rather than composing novel music. This strong content entanglement even disrupts lyrical control. Once shortcut learning occurs, the model's creative capabilities cannot be easily restored. Removing ICL data and continuing training on SPC alone fails to resolve the issue—without reference audio, the model struggles to generate meaningful outputs, exhibiting poor musicality.

To address this, we introduce a **delayed activation strategy**. We introduce a small amount of ICL data (∼10B tokens) only during the annealing phase, ensuring no ICL data is used beforehand. This

strategy facilitates **disentangled** control between text and reference audio. For instance, using a Japanese city pop track with a female vocal as a reference, the model can transform the lyrics into English while preserving the same vocalist and genre, or even generate a male rap version of it.

## 2.5 TOKENIZATION AND STAGE-2 LM

**Tokenization.** Our text vocabulary relies on the LLaMA tokenizer, which handles instructions, genres, and lyrics. We expand the vocabulary to support audio tokens. We use `X-Codec` (Ye et al., 2024) as our audio tokenizer. It combines semantic and acoustic information in a fused token, facilitating faster convergence. A lightweight upsampling module is also applied. More details are in Appendix C.

**Stage-2 Residual Modeling.** Given Stage-1's tokens (codebook-0), Stage-2 uses an AR LM to *jointly* predict all $K = 8$ codebooks (0–7) on a strictly time-aligned stream, which first reads the entire codebook-0, then emits per-frame 8-tuples to align semantics with residual detail. The model is trained with teacher forcing; at inference, codebook-0 is clamped to Stage-1 and only residual codebooks $1-7$ are generated, preserving alignment while refining audio. More details in Appendix D.

## 3 EXPERIMENTS

### 3.1 DETAILED SETUP

**Pre-training.** During pre-training, we employ multitask learning, including TTS, lyrics-to-song, and unconditional music generation tasks to jointly develop vocal and instrumental modeling. The final Stage-1 LM has 7B parameters, and the Stage-2 LM has 2B. 70k hours of speech and 650k hours of creative commons license music mined from the web are used. Prior to annealing, the data mixture is set at *Conditional : Unconditional = 3 : 1* and *Music : Speech = 10 : 1*. During annealing, only SPC and ICL data are used, maintaining a ratio of *SPC : ICL = 2 : 1*.

Most Stage-1 experiments use a 0.5B model with a 100B token budget. For scaling, we increase the token budget to 500B and scale models to 0.5B, 2B, and 7B parameters. The 7B model is further trained on 1.75T tokens with 16K context, followed by a 40B-token annealing phase. Stage-2 experiments use 2T tokens and 8K context. We maintain a global batch size of 768. A maximum LR of 3e-4 with linear warm up is applied, and anneals to 3e-5 during the annealing phase.

**Test-time Strategies.** Sampling and Classifier-Free Guidance are applied to improve the good-case rate. Music ICL, which utilizes a song's chorus as a prefix, enhances musicality and stability.

### 3.2 EVALUATION PROTOCOL

**Baselines.** We compare against four widely used proprietary music-generation systems, including Suno V4, Udio, Hailuo, and Tiangong[6], which are representative of current production-grade models.

**Human Evaluation.** We conducted a human evaluation involving 40 researchers, including 12 experts in Speech/Music AI[7] and 7 trained musicians. None of the evaluators participated in model training, ensuring objectivity. Following prior studies (Donahue et al., 2023; Qu et al., 2024; Yuan et al., 2024), we adopted an A/B test format. See Appendix F for detailed protocol.

**Automatic Evaluation.** We also report automatic evaluation metrics, including Kullback–Leibler (KL) divergence for measuring distributional differences in generated audio features using `audioldm_eval`[8], Frechet audio distance (FAD) (Kilgour et al., 2019) for assessing audio quality and realism (also via `audioldm_eval`), Audiobox-Aesthetic (Tjandra et al., 2025) for capturing perceived musical aesthetics based on production quality (PQ), production complexity (PC), content enjoyment (CE), and content usefulness (CU) using a neural audio embedding model, CLAP score[9] and CLaMP 3 score (Wu et al., 2025)[10] to measure semantic alignment between text prompts

---

[6]`https://suno.com` , `https://www.udio.com` , `https://hailuoai.com/music` , `https://www.tiangong.cn/music`

[7]Worked on text-to-speech, text-to-music, singing voice synthesis.

[8]`https://github.com/haoheliu/audioldm_eval`

[9]`https://github.com/Stability-AI/stable-audio-metrics`

[10]`https://github.com/sanderwood/clamp3`

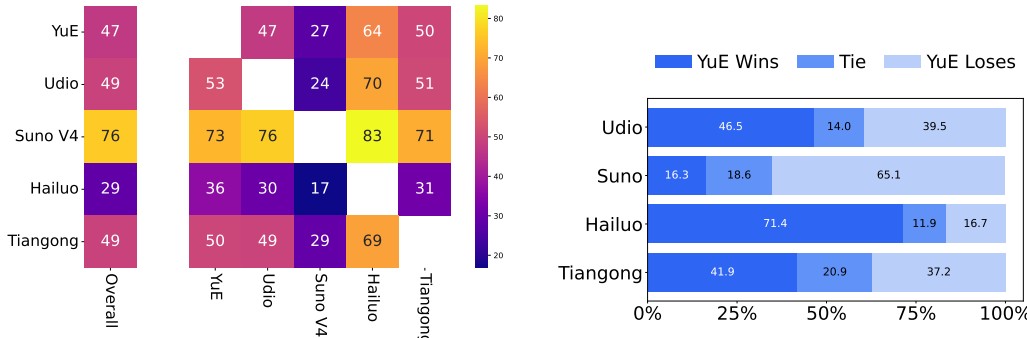

Figure 3: Human evaluation comparing YuE to 4 **proprietary systems**. YuE matches two of it (Tiangong, Udio) and outperforms one (Hailuo). **Left**: Average human preference on all aspects. Each cell (y, x) denotes the win rate of the system on the y-axis against the system on the x-axis. The leftmost "Overall" column shows the aggregate win rate against all other systems. Brighter colors / larger numbers indicate higher preference. **Right**: win-tie-loss on musicality.

and audio outputs, vocal agility quantifying song-level vocal range and flexibility (pitch estimated with RMVPE[11], applying 40ms note filtering and human verification), and generation duration as a practical measure of song-level audio modeling capability.

## 4 MAIN RESULTS

**Human Evaluation.** As shown in Fig. 3, YuE is competitive with four proprietary systems on both average human preference[12] and musicality. It surpasses Hailuo by a clear margin, performs on par with Tiangong and Udio, but trails Suno V4 (the strongest system in our comparison). Focusing on musicality, YuE shows roughly balanced win–loss rates against Tiangong and Udio, decisively outperforms Hailuo, yet remains behind Suno V4. Overall, while proprietary models still lead, YuE narrows the gap and offers a strong open-source alternative for music generation.

**Vocal Agility.** As shown in Fig. 4, the distribution of song-level vocal ranges across different systems reveals notable variations in vocal agility. Higher values indicate greater vocal expressiveness. Among the models, YuE demonstrates one of the widest vocal ranges (medium $\sim= 27$ semitones), closely matching top-performing closed-source systems like Suno V4. This suggests that YuE is capable of generating diverse and dynamic vocal performances. In contrast, models like Hailuo and Tiangong show a more constrained vocal range (a medium number around 20 semitones), indicating potential limitations in expressiveness. These findings highlight YuE's strength in producing vocally rich and varied song compositions.

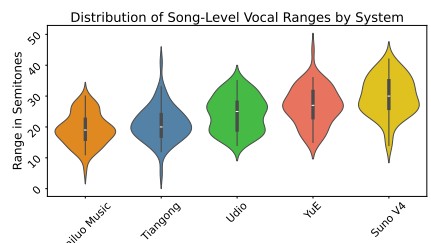

Figure 4: Song-level vocal range on different systems. Higher values indicates better vocal agility, e.g. range=12 means the vocal only span through an octave in a given song. YuE's vocal range is among the top close-source systems.

**Duration.** The distribution of generated song durations across different systems reveals substantial variation in length constraints as demonstrated by Fig. 5. YuE produces the longest audio, with a significantly wider duration range compared to all other models, demonstrating its ability to generate full-length songs beyond typical AI-generated clips. SunoV4 and Tiangong also generate relatively long audio. In contrast, Hailuo Music show the most restricted durations, suggesting limitations in modeling long-term musical structure. These results

---

[11]https://github.com/yxlllc/RMVPE

[12]Averaged win rate across all aspects.

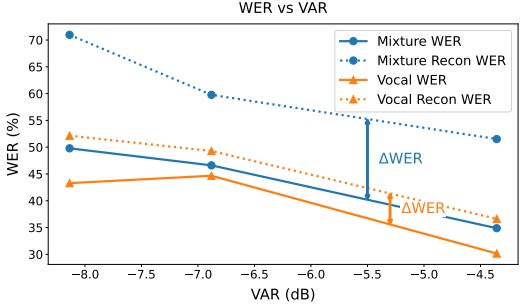

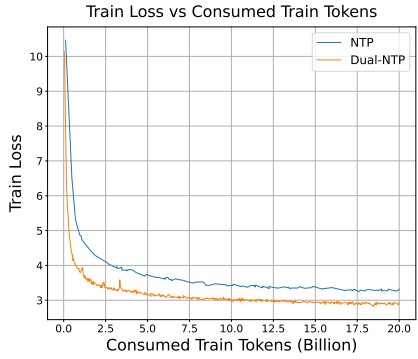

Figure 6: Comparison of WER-VAR plot for mixture and vocal tracks, including their tokenizer reconstructions, over 1K samples.

Figure 7: Training Loss over Consumed Train Tokens for NTP and Dual-NTP

highlight YuE's advantage in handling extended temporal dependencies, making it more suitable for full-song generation.

**Model-based Evaluation.** Table 1 shows model-based results, including distribution metrics KL and FAD, aesthetics metrics, and audio-text alignment scores such as CLAP score (Wu et al., 2023) and CLaMP 3 score (Wu et al., 2025). YuE outperforms others in KL divergence (0.372) and performs competitively in FAD (1.624), showing strong audio quality and distribution matching. It also scores well on content-based metrics, with scores close to top systems like SunoV4 and Tiangong, indicating strong audio aesthetics and usability. Regarding alignment metrics, YuE achieves the highest score with CLaMP 3 (0.240) but has a lower CLAP score (0.118). These

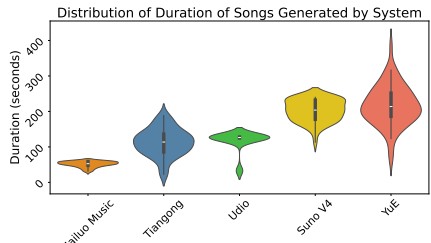

Figure 5: Duration range on different systems. YuE generates the longest audio.

discrepancies highlight potential limitations of the CLAP score in accurately capturing human perceptions of controllability, possibly due to differences in pretraining data and modeling strategies[13]. **Note that not all metrics align well with human perception**. A detailed discussion of correlation between each metric and human evaluation is provided in Appendix H.

Table 1: Comparison of various music generation models across multiple metrics.

| Metric | Distrib. Match | | Content Based | | | | Alignment | |
|---|---|---|---|---|---|---|---|---|
| | KL↓ | FAD↓ | CE↑ | CU↑ | PC↑ | PQ↑ | CLAP↑ | CLaMP 3↑ |
| Hailuo | 0.756 | 2.080 | 7.350 | 7.737 | **6.793** | 8.132 | 0.265 | 0.106 |
| SunoV4 | 0.620 | 1.544 | **7.474** | **7.813** | 6.601 | 8.120 | 0.265 | 0.160 |
| Tiangong | 0.708 | 2.547 | 7.421 | 7.766 | 6.060 | **8.220** | 0.244 | 0.114 |
| Udio | 0.503 | **1.222** | 7.112 | 7.520 | 6.626 | 7.803 | **0.310** | 0.156 |
| YuE | **0.372** | 1.624 | 7.115 | 7.543 | 6.280 | 7.894 | 0.118 | **0.240** |

## 5 ANALYSIS AND ABLATIONS

**Effect of Source Separation Prior and Dual-NTP.** We define a metric called the **vocal-to-accompaniment Ratio (VAR)**, to quantify the effect of track-wise energy distribution on linguistic information loss. Let $v(n)$ denote the vocal signal and $a(n)$ denote the accompaniment signal, over

---

[13]CLaMP 3 is a more recent model compared to CLAP, showing improved results in representation quality and music retrieval tasks due to extensive web-scale pretraining. In contrast, CLAP may be limited by its exposure to singing/music content during training, causing discrepancies in evaluating certain music types..

$n = 1, 2, \ldots, N$. We compute VAR (in dB) as follows:

$$\text{VAR} = 10 \log_{10} \sum_{n=1}^{N} v^2(n) - 10 \log_{10} \sum_{n=1}^{N} a^2(n), \quad (6)$$

where higher VAR values indicate greater prominence of vocals relative to accompaniment, while lower VAR values suggest accompaniment dominance.

Figure 6 illustrates the WER-VAR relationship for mixture and vocal tracks across 1K samples, including tokenizer reconstructions. Though original vocal and mixture tracks exhibit similar absolute WER (solid blue and orange lines), mixture track reconstruction significantly increases WER (solid vs. dotted blue lines), especially as VAR declines, widening the gap ($\Delta$WER). A 20%+ $\Delta$WER is observed around -8.0 dB VAR. In contrast, vocal tracks maintain low WER and smaller $\Delta$WER (the worst case is 10%- around -8.0dB VAR), indicating resilience of source separation priors to VAR degradation and reconstruction information loss.

We also perform an ablation study comparing Dual-NTP and standard NTP. Figure 7 presents training loss curves of two 0.5B LMs trained with identical data and computational budgets (20B tokens). Dual-NTP shows a substantial reduction in loss (approximately 0.4 lower) compared to NTP, confirming its faster convergence. These analyses underscore the effectiveness of incorporating source separation priors with Dual-NTP into the song modeling task.

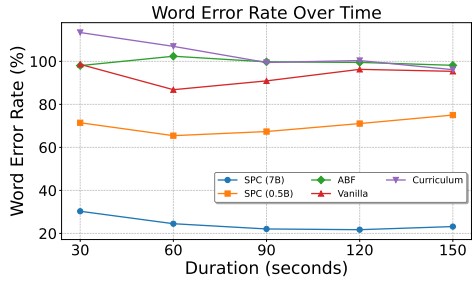

Figure 8: WER over time. Both SPC and model scaling enhance lyrics-following capability.

**Effect of Lyrics-following Capabilities with SPC.** The analysis in Figure 8 examines an ablation setting involving a 0.5B LM, which was initially pretrained on a default mixture dataset[14] comprising 500B tokens and subsequently finetuned on the corresponding lyrics data for an additional 200B tokens using the specified methods: **Vanilla**, **Curriculum**, and **ABF**, and our proposed **SPC**. Additionally, we include results from the YuE-7B checkpoint to illustrate the performance gains achievable through scaling. Specifically,**Vanilla** refers to text prepend conditioning, where the model is trained with prepended lyrics as input for conditioning. **Curriculum** involves gradually increasing the text prepend data with progressively longer durations (e.g., 30s, 60s, 90s, etc.), aiming to improve the model's ability to follow lyrics over time. **ABF** (Xiong et al., 2023) refers to adjusting the rope base frequency from 10k to 100k during finetuning to explore its effect on lyrics-following performance. In Figure 8, the WER over time is estimated using a fine-tuned Whisper model, with measurements recorded every 30 seconds up to 150 seconds. Overall, the proposed SPC method achieves consistently superior performance across all evaluated time intervals (30s to 150s). Scaling the model to 7B parameters demonstrates substantial improvements, reducing the WER from approximately 70% at 0.5B parameters to around 20%[15]. In contrast, Vanilla, Curriculum, and ABF methods exhibit substantially worse WER, indicating a limited capability in maintaining lyrical coherence. Through manual inspection, we identified that the primary reason for failure in Vanilla and Curriculum was their tendency to generate instrumental preludes, causing the onset of singing to drift far from the original prepended lyrics condition.

**Effect of Scaling.** We investigate the impact of model scaling on musicality and lyrics-following capabilities. We compared checkpoints at 0.5B, 2B, and 7B scales. While the 0.5B and 2B models were trained with a limited budget of 500B tokens (in 16K context), the 7B model underwent complete scaling with a significantly larger 1.75T token budget using the full training dataset. In Figure 9a, human evaluation demonstrates a clear improvement trend in both musicality and lyrics-following as model scale and training budget increase. Notably, the 7B model exhibits substantial enhancements, indicating that increased parameter counts and extensive training significantly boost the model's foundational creativity and compositional quality. These results confirm that scaling plays a crucial role in achieving higher musicality and improved lyric adherence.

---

[14]A mixture of speech and music. Text transcripts are in prepend format.

[15]Note that 20% can be considered a relatively low number. Refer to the GT WER-to-VAR plot in Figure 6.

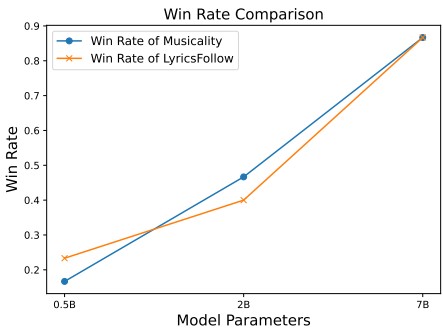

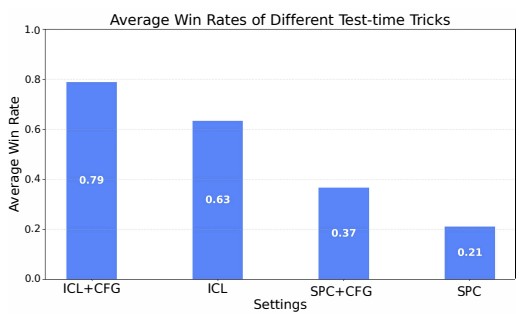

(a) Win rates for Musicality and Lyrics-following across model scales in pairwise A/B tests.

(b) Win rates for musicality.

Figure 9: Win Rate Statistics.

**Effect of Test-time Tricks.** Figure 9b presents human preference win rates for musicality obtained through A/B testing across different inference settings using YuE-7B checkpoints. Results clearly demonstrate that ICL-based methods outperform SPC-based methods significantly: ICL achieves a win rate of 0.63 compared to only 0.21 for SPC. Incorporating CFG further enhances these methods; specifically, ICL+CFG obtains the highest win rate (0.79), substantially exceeding both ICL alone and the SPC-based configurations. This performance advantage stems from the strong conditioning ability of ICL, which restricts the decoded token space to a musically favorable subspace guided by the provided human-generated music prompt. CFG similarly strengthens this conditioning by amplifying the influence of the text condition on next-token logits, making generated outputs more closely aligned with the intended prompt-guided subspace and thus further improving musicality.

**Memorization Test.** To assess if YuE's ICL mode (30 s audio prompt + lyrics) reproduces training material, we compute a melody-aware cosine similarity with ByteCover2 (Du et al., 2022) between $N$=1200 matched **Ref** (training)–**Gen** (ICL outputs) pairs, analysing the top-1% tail as in prior work (Agostinelli et al., 2023; Yuan et al., 2024). For context, we report the same metric on GTZAN (genre-level pairs) and Covers80 (known duplicates). As shown in Fig. 10, *Ref–Gen* similarities are far below Covers80 and comparable to GTZAN. While occasional short motifs (*e.g.*, percussive loops) appear, we find no evidence of wholesale copying; YuE primarily recombines learned patterns to produce novel content.

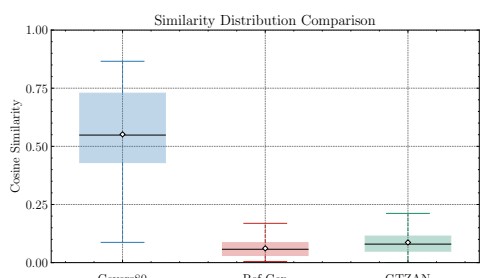

Figure 10: Box-plot comparison of cosine similarity across three scenarios: Covers80, *Ref-Gen* (our training vs. generated sets), and GTZAN. The black bar denotes the median, and the diamond denotes the mean.

**Emergent Abilities.** Scaling YuE substantially improves generation quality and reveals emergent behaviors: advanced vocal techniques (e.g., vibrato, bel canto, death growl), spontaneous musicality (scat continuation, a cappella harmonies, instrumental fills), global style fusion, and code switching. The model also achieves realistic singing voice cloning and versatile style transfer across languages, genres, and timbres while preserving lyrical/melodic structure and prosody, illustrated in our demo[16].

## CONCLUSION

We introduced YuE, an open-source foundation model family for long-form lyrics-to-song generation. Leveraging large-scale data, Dual-NTP, SPC, and a redesigned ICL framework, YuE generates coherent, expressive songs with detailed musical structure. Experiments demonstrate that YuE matches or exceeds commercial systems in musicality, controllability, and cross-lingual lyrics following, and achieves competitive results on standard music understanding benchmarks.

---

[16]https://map-yue.github.io/

## CONTRIBUTIONS AND ACKNOWLEDGMENTS

The research was supported in part by Early Career Scheme (ECS-HKUST22201322), Theme-based Research Scheme under Grant T45-205/21-N from Hong Kong RGC, and Generative AI Research and Development Centre from InnoHK.

**Core Contributors**

Ruibin Yuan, Lead, Pre-train, Data, Eval
*HKUST, Moonshot.ai, MAP, ryuanab@connect.ust.hk*

Hanfeng Lin, Pre-train, Data, Eval, Inference
*HKUST, MAP, hanfengl@ust.hk*

Shuyue Guo, Pre-train, Demo
*MAP*

Ge Zhang, Pre-train
*MAP, gezhang@umich.edu*

Jiahao Pan, Pre-train, Eval, Data
*HKUST, MAP, fengshicherish@gmail.com*

**Contributors**

Yongyi Zang, Upsampler, Eval
*Independent*
Haohe Liu, Upsampler, Tokenizer, Demo
*University Of Surrey, MAP*
Yiming Liang, Eval Lead
*MAP*
Wenye Ma, Representation Learning
*MBZUAI, MAP*
Xingjian Du, Memorization Effect
*University of Rochester, MAP*
Xinrun Du, Pre-train
*MAP*
Zhen Ye, Tokenizer
*HKUST*
Tianyu Zheng, Pre-train
*MAP*
Zhengxuan Jiang, Inference
*MAP*
Yinghao Ma, Eval
*MAP, Queen Mary University of London*
Minghao Liu, Eval, Data
*2077AI, MAP*
Zeyue Tian, Eval
*HKUST, MAP*
Ziya Zhou, Eval, Data
*HKUST, MAP*
Liumeng Xue, Eval, Data
*HKUST, MAP*
Xingwei Qu, Pre-train, Eval
*MAP*
Yizhi Li, Eval
*MAP, University of Manchester*
Shangda Wu, Eval
*Central Conservatory of Music, MAP*
Tianhao Shen, Eval, Inference
*MAP*

Ziyang Ma, Eval
*MAP, SJTU, NTU*
Jun Zhan, Eval
*Fudan University*
Chunhui Wang, Eval, Pre-train
*Geely*
Yatian Wang, Eval
*HKUST*
Xiaowei Chi, Eval
*HKUST*
Xinyue Zhang, Eval
*HKUST*
Zhenzhu Yang, Eval
*HKUST*
Xiangzhou Wang, Eval
*MAP*
Shansong Liu, Eval
*Meituan*
Lingrui Mei, Eval
*Meituan*
Peng Li, Eval
*HKUST*
Junjie Wang, Eval
*Tsinghua University*
Jianwei Yu, Data, Inference
*Moonshot.ai*
Guojian Pang, Inference
*MAP*
Xu Li, Eval
*Xiaohongshu*
Zihao Wang, Data
*Zhejiang University, Carnegie Mellon University*

**Academic Advisors**

Xiaohuan Zhou
*MAP*
Lijun Yu
*Carnegie Mellon University*
Emmanouil Benetos
*Queen Mary University of London, MAP*
Yong Chen
*Geely*
Chenghua Lin
*University of Manchester, MAP*
Xie Chen
*Shanghai Jiao Tong University*
Gus Xia
*MBZUAI, MAP*
Zhaoxiang Zhang
*Chinese Academy of Sciences*
Chao Zhang

*Tsinghua University*
Wenhu Chen
*University of Waterloo, MAP*
Xinyu Zhou
*Moonshot.ai*
Xipeng Qiu
*Fudan University*
Roger Dannenberg
*Carnegie Mellon University, MAP*

Jian Yang
*MAP, jiaya@buaa.edu.cn*

Wenhao Huang
*MAP, rubio8741@gmail.com*

Wei Xue
*HKUST, weixue@ust.hk*

Xu Tan
*Moonshot.ai, MAP, tanxu2012@gmail.com*

**Correspondence (Alphabetical Order)**
Jiaheng Liu
*Nanjing University, MAP, 13121221227@163.com*

Yike Guo
*HKUST, yikeguo@ust.hk*

ETHICS STATEMENT

The authors have read and adhered to the ICLR Code of Ethics. We have taken proactive steps to address ethical considerations throughout this research, including responsible data sourcing under Creative Commons licenses, analysis of model fairness and bias across different languages and genres, and mitigation strategies for potential misuse, such as the generation of "deepfake music." Our open-source release includes binding use terms to prevent misuse. This study was reviewed and approved by the Human and Artefacts Research Ethics Committee under protocol HREP-2023-0230. A comprehensive discussion of our ethical considerations, including data handling, bias analysis, and risk mitigation, is provided in the Appendix B.

REPRODUCIBILITY STATEMENT

To ensure the reproducibility of our research, we has released our source code on GitHub[17] and pre-trained model checkpoints on HuggingFace[18]. The Appendix C and  D further details the tokenization and model architecture, while Section 3 provides hyperparameters and the computational environment required to replicate our experiments. We are confident that these resources will enable the research community to verify our findings and build upon our work.

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

## A  THE USE OF LARGE LANGUAGE MODELS (LLMS)

During the preparation of this manuscript, we utilized a Large Language Model (LLM) as a general-purpose writing assistant. Its role was confined to proofreading for grammatical errors, correcting typos, and improving sentence structure for better readability. The LLM did not contribute to the core research ideas, experimental methodology, or the substantive content of the paper.

## B  ETHICS AND RESPONSIBILITY

**Data Sources and Licensing.**  We are committed to responsible data collection. The primary sources for our training data are as follows:

- **Audio Data**: The audio tracks were sourced from an online streaming platform. To respect creators' rights, we specifically filtered for and downloaded content licensed under Creative Commons (CC), which permits reuse.

- **Lyrics Data**: The corresponding lyrics for the audio tracks were programmatically retrieved using Google Search.

- **Opt-Out and Filtering**: During data collection, we implemented a string matching filtering process to automatically exclude any data that carried an explicit copyright declaration or restrictive terms of use. This serves as our primary opt-out mechanism.

While we have taken these steps, we acknowledge the inherent challenges in verifying the copyright status for every item in a web-scale dataset.

**Language Distribution.**  The language of the lyrics was estimated using an external language identification model. The distribution is skewed towards high-resource languages, with English being the most prevalent.

Table 2: Language distribution of lyrics (estimated).

| Language ID | Percentage |
|---|---|
| en | 60.68% |
| zh | 22.83% |
| es | 6.07% |
| fr | 1.49% |
| it | 1.11% |
| ja | 1.00% |
| ru | 0.98% |
| pt | 0.96% |
| de | 0.89% |
| ko | 0.72% |

**Genre Distribution.**  Top genre labels were extracted from available metadata.

Table 3: Genre distribution.

| Genre | Percentage |
|---|---|
| Pop | 19.1% |
| Rock | 10.9% |
| Electronic | 8.1% |
| Classical | 7.0% |
| R&B | 4.7% |
| Rap | 2.3% |

**Artist Demographics.** We acknowledge that we do not have structured data on artist demographics. This information was not consistently available in the metadata of our data sources.

**Evaluation on Minority Languages and Genres.** We evaluate our model's performance on underrepresented groups. We curated a new test set of **1,000 world music tracks** from diverse global cultures (mostly folk/traditional music from a corresponding area). We evaluated our model, **YuE-7B**, on this dataset in single-track mode and report the perplexity (PPL) for each language below. A lower PPL indicates better performance.

The results demonstrate that our model is competent across a wide array of languages, many of which are minorities in our training data.

Table 4: Per-language perplexity on the minority-language test set (lower is better).

| Language | Mean PPL | Language | Mean PPL |
|----------|----------|----------|----------|
| fr | 25.53 | tr | 39.12 |
| ja | 27.61 | vi | 39.16 |
| da | 28.16 | bg | 39.31 |
| sw | 28.49 | hi | 39.57 |
| sl | 30.00 | lt | 40.38 |
| pt | 30.87 | de | 41.14 |
| bn | 31.09 | uk | 42.03 |
| cs | 31.74 | en | 42.87 |
| pl | 31.84 | it | 42.89 |
| th | 32.41 | ar | 43.72 |
| zh | 33.00 | ko | 44.68 |
| id | 33.15 | iw | 45.00 |
| sv | 33.39 | sk | 46.32 |
| sr | 33.52 | no | 47.79 |
| hu | 34.05 | ru | 48.80 |
| ro | 34.60 | et | 53.63 |
| fi | 34.86 | | |
| es | 35.15 | | |
| hr | 35.82 | | |
| el | 36.02 | | |
| nl | 38.51 | | |
| lv | 38.66 | | |

Notably, the model achieves relatively better performance (low PPL) on languages like French (fr), Japanese (ja), and Swahili (sw), demonstrating a promising ability to generalize beyond the majority languages.

**Plagiarism.** Ensuring ethical and responsible AI-generated music is crucial for fostering transparency, accessibility, and fair contribution to the music industry. As suggested by Ma et al. (2024), to promote accountability, we advocate for the inclusion of AI-generated / AI-assisted tags in generated content, increasing transparency for both musicians and audiences. Additionally, our memorization-effect experiments in Section 5 demonstrate that our design maintains creativity without plagiarizing, even under strong training set conditioning.

In contrast to closed-source commercial systems, our model leverages an exceptionally diverse training dataset, explicitly enriched with culturally diverse music content. This enables the model to innovate and create within niche musical styles effectively (see demos). As such, our model can serve as a **parameterized knowledge base**, contributing to the preservation and expansion of human musical artistry and cultural heritage.

**Risks and Mitigation.** Despite these positive contributions, we recognize that high-fidelity vocal and stylistic imitation enabled by YuE may inadvertently facilitate deepfake music generation, including unauthorized emulation of real artists' voices, styles, or lyrical identities. Such misuse

poses ethical concerns regarding consent, authenticity, and potential economic harm to original creators. To mitigate these risks, we propose several precautionary measures for the following work: (1) integrating imperceptible audio watermarks into generated content to trace model origin, (2) establishing clear terms of use and content attribution guidelines in downstream applications, and (3) encouraging community-driven audits of model outputs for misuse detection. We believe that proactive governance—alongside transparency, open research practices, and active engagement with stakeholders—can help balance creative empowerment with ethical responsibility.

**Current Status.**    Our open-source repository already enforces binding use terms requiring: 1) Clear labeling of AI-generated content, 2) Compliance with copyright laws, 3) Prohibition of unauthorized voice/style impersonation. 4) We are developing watermarking tools for the model.

**Ethics Committee Approval.**    This study has been reviewed and approved by the Human and Artefacts Research Ethics Committee under protocol HREP-2023-0230. The approval ensures that our research adheres to ethical guidelines in data usage, AI generation, and cultural representation. The approval remains effective until 30-Jan-2027.

## C TOKENIZATION AND AUDIO RECONSTRUCTION

### C.1 OVERVIEW

Following the design space of Borsos et al. (2023); Wang et al. (2023b), the stage-1 LM models text tokens and semantic-rich codebook-0 tokens. After investigation, we realized that the vanilla text-to-speech (TTS) / text-to-music (TTM) method performs poorly on our task, where musicality and song-level lyrics-following capability are the two key challenges.

Table 5: Special tokens and their descriptions.

| Token | Description |
|---|---|
| <EOD> | End of document |
| <SOA> | Start of audio |
| <EOA> | End of audio |
| <stage_1> | Start of Stage 1 |
| <stage_2> | Start of Stage 2 |
| <encodec32k> | Tokenizer type (Encodec 32k) |
| <xcodec> | Tokenizer type (XCodec) |
| <semanticodec> | Tokenizer type (SemantiCodec) |
| <hificodec> | Tokenizer type (HiFiCodec) |

**Text Tokenizer.** In this work, the vocabulary of the LMs contains two sections: text and audio. For the text part, we reuse LLaMA tokenizer with a size of 32000 unique BPE tokens. Instructions, genres, lyrics, structure annotations, and structure segment boundary signals are represented with text format and tokenized with BPE.

**Semantic-Acoustic Fused Codec.** For the audio vocabulary, we experimented with several open-source music and universal neural codecs. Ultimately, we adopted a semantic-acoustic fused strategy (Défossez et al., 2024; Zhang et al., 2023; Liu et al., 2024a; Ye et al., 2024). Specifically, we utilized X-Codec (Ye et al., 2024) as our off-the-shelf audio tokenizer. We employed a general-purpose version of X-Codec, trained on a mixture of 200k hours of 16 kHz audio with a ratio of music : speech : audio effects = 1 : 1 : 0.05.

The X-Codec tokenizer fuses a 100M-parameter HuBERT-based universal semantic representation into the codec latent space. It has a 50Hz frame rate, consists of 12 RVQ layers, each with a codebook size of 1024. For this study, we used only the first 8 layers, as including more layers did not yield noticeable quality improvements. Notably, codebook-0 alone captures rich semantic information such as melody and vocal content, which are critical for our task.

**Vocabulary Expansion and Special Tokens.** We expand the SentencePiece tokenizer vocabulary to support multiple audio tokenizers and special tokens. Specifically, we include Encodec-32khz-music (Défossez et al., 2022; Copet et al., 2023), HiFi-Codec-universal (Yang et al., 2023a;b), X-Codec-general (Ye et al., 2024), and Semanticodec-100tps (Liu et al., 2024a).

For special tokens, we introduce the following: <EOD> represents the end of a document, <SOA> denotes the start of audio, and <EOA> signifies the end of audio. Additionally, stage indicators, <stage_1> and <stage_2>, mark the beginning of Stage 1 and Stage 2 tokens, respectively. Tokenizer type indicators specify the corresponding tokenizer types, which are inserted between <SOA> and the actual audio token IDs. Note that stage indicators are only used in residual modeling and positioned between <SOA> and the tokenizer type indicator.

**Light-weight Upsampling Module.** To achieve better perceptual audio quality, we upsample the reconstructed 16kHz audio to 44.1kHz. For this, we utilize a light-weight upsampling vocoder adapting Vocos (Siuzdak, 2023) to predict the higher-frequency components. To enhance the robustness of the upsampler, we apply codebook dropout randomly and introduce a small amount of Gaussian noise during training.

### C.2 COMPARISON OF AUDIO TOKENIZERS

In preliminary experiments on a 130k-hour subset of diverse music data, we conducted a qualitative analysis of four popular audio tokenizers, specifically focusing on acoustic tokens and fused semantic-acoustic tokens (see Table 6). Separate semantic and acoustic tokenizers would require retraining and thus were beyond the scope of this study, reserved for future work.

Table 6: Qualitative comparison of different codec types based on reconstruction quality, LM convergence, and invalid probability. Invalid probability refers to the likelihood of generating noise or silence segments during LM token synthesis.

| Type | Codec | Reconstruction | LM Converge | Invalid Prob. |
|------|-------|----------------|-------------|---------------|
| Acoustic | Encodec32k | Good | No | All |
| Acoustic | HiFiCodec | Good | No | All |
| Semantic + Acoustic | Semanticodec | Fair | Yes | High |
| Semantic + Acoustic | X-Codec | Fair | Yes | Low |

Acoustic tokenizers, including Encodec32k and HiFiCodec, exhibited decent reconstruction quality. However, their learned tokens proved challenging for LMs to converge due to the complexity and variability inherent in our in-the-wild dataset. Training a 0.5B LM with acoustic tokens consistently failed to converge, resulting primarily in invalid outputs characterized by noise or silence. Although prior studies indicated Encodec32k has been successfully applied to TTM (Copet et al., 2023), even scaling the LM to 7B and extending training up to 1 trillion tokens on our data yielded only intermittent success, with outputs still dominated by noise.

In contrast, tokenizers integrating semantic and acoustic features (Semanticodec, X-Codec) demonstrated significantly better convergence, largely due to the stable clustering provided by SSL encoders. This stability facilitated successful LM training at the 0.5B scale. However, the stable clustering slightly compromised acoustic dynamics, causing only fair reconstruction quality. We further identified a critical alignment flaw in Semanticodec related to AudioMAE's patch-based mechanism, where misalignment of one token propagated errors throughout reconstruction. X-Codec, using Hubert-derived semantics, avoided this issue and maintained lower invalid generation probability.

# D  STAGE-2: RESIDUAL MODELING

As shown in Figure 11, after Stage-1 yields coarse semantic tokens (codebook-0), Stage-2 refines the audio with additional codebooks $1, 2, \ldots, 7$. Denote the total number of codebooks by $K = 8$ (indexed from 0 to 7). Although codebook-0 is already produced by Stage-1, we train Stage-2 to predict *all* codebooks $\{0, 1, \ldots, 7\}$ jointly in a single autoregressive framework. This design ensures that the model has a unified view of both the high-level structure (codebook-0) and the residual details (codebooks 1–7).

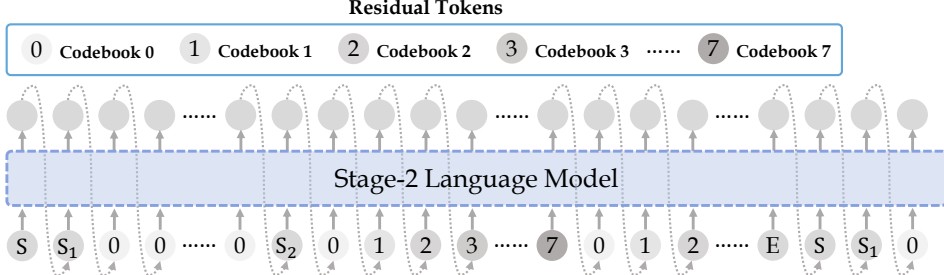

Figure 11: Stage-2 Framework of YuE. $S$: `<SOA>`, $S$=`<SOA>`, $E$=`<EOA>`, $S_i$=`<stage_i>`.

**Architecture Overview.** Let $\mathbf{x}_{1:T}^{(0)} = (x_1^{(0)}, \ldots, x_T^{(0)})$ be the Stage-1 codebook-0 tokens for $T$ frames. In Stage-2, we introduce additional codebooks, collectively denoted by

$$\mathbf{x}_{1:T}^{(1:7)} \;=\; \big(x_1^{(1)}, \ldots, x_1^{(7)}; \; \ldots; \; x_T^{(1)}, \ldots, x_T^{(7)}\big).$$

For training, we treat the output space as $\mathbf{x}_{1:T}^{(0:7)}$, i.e., each timestep $t$ has a tuple

$$\mathbf{x}_t^{(0:7)} = \big(x_t^{(0)}, x_t^{(1)}, \ldots, x_t^{(7)}\big).$$

Although codebook-0 tokens are the same as those from Stage-1, they are included in the training target so the model learns to predict them as well, thus capturing complete frame-level dependencies across all codebooks.

**Aligned Autoregressive Factorization.** We maintain a strictly time-aligned factorization:

$$p\Big(\mathbf{x}_{1:T}^{(0:7)}\Big) \;=\; \prod_{t=1}^{T} p\Big(\mathbf{x}_t^{(0:7)} \,\Big|\, \mathbf{x}_{<t}^{(0:7)}\Big). \tag{7}$$

This ensures that at each frame $t$, the model conditions on all previously generated tokens across *all* codebooks, while still maintaining frame alignment with codebook-0.

**Cross-Conditioning.** During training, we organize the sequence as:

$$\Big[\underbrace{x_1^{(0)}, \ldots, x_T^{(0)}}_{\text{all codebook-0 first}}, \underbrace{x_1^{(0)}, x_1^{(1)}, \ldots, x_1^{(7)}, \; x_2^{(0)}, x_2^{(1)}, \ldots, x_2^{(7)}, \; \ldots, \; x_T^{(0)}, x_T^{(1)}, \ldots, x_T^{(7)}}_{\text{blocks of 0-7 per frame}}\Big].$$

That is, the first segment is only the codebook-0 tokens, followed by repeated 8-token blocks $\{0, 1, \ldots, 7\}$ for each frame. We apply standard teacher forcing on this extended sequence and minimize

$$\mathcal{L}_{\text{Stage2}} \;=\; -\sum_{t=1}^{T} \log p\Big(\mathbf{x}_t^{(0:7)} \,\Big|\, \mathbf{x}_{<t}^{(0:7)}\Big).$$

By placing all codebook-0 tokens at the beginning, the model is guaranteed to "see" the entire semantic structure before it encounters any mixed (0–7) blocks. This allows the model to plan the later residuals by attending to a complete semantic outline from Stage-1.

**Inference.** At test time, codebook-0 tokens $\mathbf{x}_{1:T}^{(0)}$ come from Stage-1 and are treated as fixed (i.e., clamped). Even though the model is trained to predict codebook-0 as part of the joint sequence,

during inference we replace any predicted codebook-0 tokens with the Stage-1 output. Consequently, the only "free" outputs in the autoregressive generation are the residual codebooks $\mathbf{x}_{1:T}^{(1:7)}$. This ensures the sequence alignment.

**Implementation.** Our model is a 2B-parameter Transformer with an 8K-token context window, trained on consecutive 6-second single-track segments. It employs a shared acoustic codebook space to model various audio types, including speech, vocals, instrumentals, and mixtures.

# E  LINGUISTIC INFORMATION LOSS AFTER TOKENIZATION

We quantify **L**inguistic information **L**oss **A**fter **T**okenization (**LLAT**) using delta **W**ord **E**rror **R**ate ($\Delta$WER), defined as $\Delta\text{WER} = \text{WER}_{\text{recon}} - \text{WER}_{\text{ori}}$, where $\text{WER}_{\text{recon}}$ and $\text{WER}_{\text{ori}}$ are estimated by a fine-tuned Whisper[19] model on tokenizer-reconstructed[20] and original mixture audio, respectively. Figure 12 illustrates the relationship between $\Delta$WER and music genre (hip-hop, pop, metal) using 1k sampled tracks. An upward trend is evident, with metal exhibiting the highest LLAT followed by pop and hip-hop, indicating greater modeling difficulty in acoustically dense genres. Vocal-only tracks consistently achieve lower $\Delta$WER compared to mixtures, indicating lower LLAT after source separation.

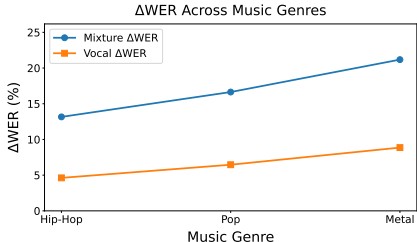

Figure 12: $\Delta$WER across different music genres for mixture / vocal-only tracks. $\Delta$WER $\propto$ LLAT.

---

[19] A Whisper V3 checkpoint fine-tuned on an internal song dataset with manual transcription.

[20] We use `X-Codec` as our tokenizer. We use 8-codebook in this LLAT experiment. See more discussion in Appendix C and  C.2.

# F SUBJECTIVE EVALUATION

## F.1 EVALUATION METHODS

In this subjective evaluation experiment, annotators were required to perform pairwise comparative evaluations of music generation outputs from multiple models. Each test unit comprised two distinct musical pieces generated by different models. Following complete playback of both samples, annotators conducted binary comparative selections (options: Superiority of A, Superiority of B, or Equivalence between A and B) across predefined evaluation dimensions. Mandatory preference judgments were enforced for each dimensional criterion, with explicit instructions to minimize the frequency of selecting the equivalence option. The evaluation protocol incorporated a double-blind procedure with randomized presentation order of audio pairs to mitigate potential ordering effects.

In the main experiments in Section 4, each model generated 42 full-length songs based on a diverse set of English prompts specifying genre, instruments, emotion, lyrics, and tempo. These prompts utilized real lyrics that were rewritten by GPT and paired with corresponding 30s chorus segments as reference audio. Evaluators blindly compared pairs of music pieces produced by two different systems.

For ablation studies in Section 5, unless otherwise specified, we utilize a set of 15 GPT-generated English prompts (see Appendix I). Each study undergoes small-scale A/B testing, with inference performed twice per prompt, resulting in a total of 30 samples per setting.

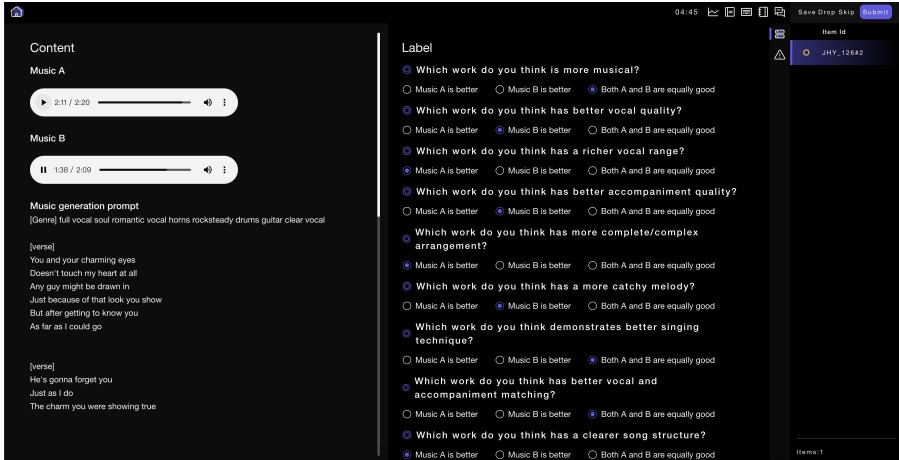

Figure 13: Subjective evaluation platform.

## F.2 EVALUATION PROTOCOL

1) **Overall Musicality**
   **Definition:** The musical artistic value and professionalism demonstrated by the work as a whole, reflecting whether it approaches the creative level of professional musicians or composers.
   **Evaluation Criteria:** Smoothness of the melody, complexity and rationality of the harmony, precision and rhythmic flow, and the artistic and creative qualities of the overall arrangement.

2) **Vocal Quality (VocalQual)**
   **Definition:** The acoustic quality of vocal performance in the work.
   **Evaluation Criteria:** Pitch, rhythmic stability, naturalness of vocal timbre (resembling human singing), fullness and warmth of timbre, degree of mechanical or distorted sound, clarity of vocals, and richness in capturing delicate emotional expressions (e.g., variations in breath control, articulation precision, emotional conveyance).

3) **Accompaniment Quality (AccompQual)**
   **Definition:** The acoustic quality of the instrumental accompaniment in the work.
   **Evaluation Criteria:** Realism and authenticity of instrumental timbres, dynamic variation

and detail richness in instrumental expression (e.g., subtlety in guitar plucking or percussion dynamics).

4) **Music Arrangement Complexity (MusicArr)**
**Definition:** The layering, coherence, balance, and creativity of the musical arrangement in the work.
**Evaluation Criteria:** Clarity of arrangement layers, coordination and interplay between instruments, balance of accompaniment within the overall audio track (e.g., appropriate volume and frequency distribution), fullness of low frequencies, brightness of high frequencies, diversity of arrangement elements (e.g., harmony, melodic lines, rhythm patterns across multiple dimensions), creativity, and variation and emotional progression between sections.

5) **Melodic Attractiveness (MelodicAttrac)**
**Definition:** The memorability, accessibility, and resonance-inducing capability of the melody.
**Evaluation Criteria:** Ease of memorization and singability, catchiness, emotional resonance, and repeated hooks or memorable elements, especially in the chorus.

6) **Vocal-Accompaniment Compatibility (VocalAccompComp)**
**Definition:** The consistency and compatibility between vocal melodies and instrumental accompaniment in terms of musical style, modality, harmony, and rhythm.
**Evaluation Criteria:** Compatibility of vocal melodies and accompaniment in modality, harmony, and rhythm, and absence of dissonance or conflict.

7) **Song Structure Clarity (SongStruct)**
**Definition:** The logical coherence and sectional distinctiveness of the overall song structure.
**Evaluation Criteria:** Clarity of the song's structure (e.g., differentiation among verses, choruses, and interludes), naturalness of transitions between sections, and structural completeness.

### F.3 Conditional Evaluation Dimension and Definitions

8) **Lyrics Following (LyricFollow),**
**Definition:** The accuracy of AI-generated vocals in performing the lyrics specified in the prompt.
**Evaluation Criteria:** Accuracy of lyric delivery (whether the specified lyrics are correctly performed), clarity of pronunciation (whether the lyrics are intelligible), alignment of lyrics rhythm with the musical beat, and naturalness and correctness of multilingual lyric transitions and pronunciations.

9) **Genre Controllability (GenCtrl)**
**Definition:** The degree to which the generated music accurately reflects the musical genre specified in the prompt.
**Evaluation Criteria:** Accuracy of musical genre characteristics (whether the generated music aligns with the features of the genre specified in the prompt, such as jazz, pop, classical, rock, etc.).

10) **Instrument and Vocal Configuration Controllability (InstrCtrl)**
**Definition:** The extent to which the generated music adheres to the instrument and vocal configuration specified in the prompt.
**Evaluation Criteria:** Matching of instrument and vocal configuration (whether the generated music follows the specifications in the prompt, such as piano, guitar, male or female vocals, choir, etc.).

11) **Emotional Expressiveness (EmoCtrl)**
**Definition:** The accuracy and impact of emotional expression in the generated music, as specified in the prompt.
**Evaluation Criteria:** Alignment of musical emotions with the emotional description in the prompt (e.g., passionate, sorrowful, cheerful).

12) **Tempo and Rhythm (Tempo/RhyCtrl)**
**Definition:** The congruence of the music's tempo (BPM) and rhythm with the requirements specified in the prompt.
**Evaluation Criteria:** Consistency of generated music tempo (BPM) with the tempo specified in the prompt, and adherence to the required rhythmic patterns.

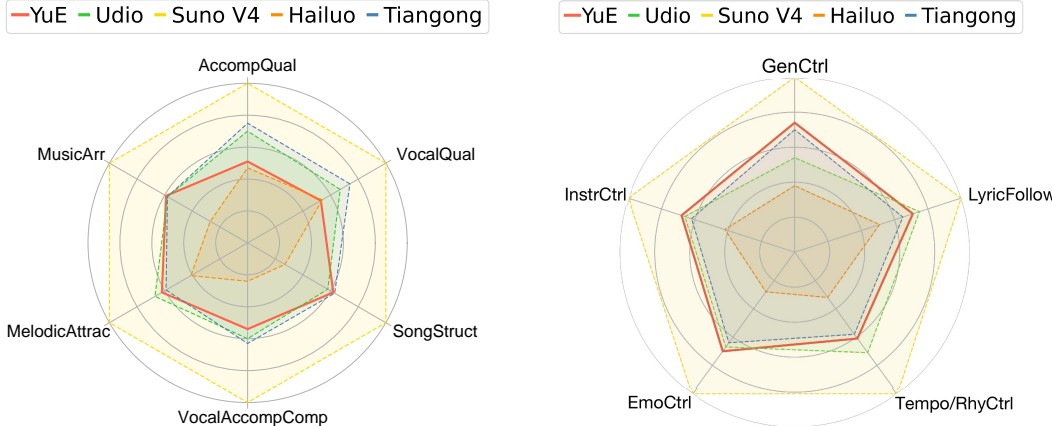

Figure 14: Normalized human preference on different music aspects. Left: scores across 6 musical aspects; Right: performance on 5 types of control.

## G   DETAILED COMPARISON WITH PROPRIETARY SYSTEMS ON SUBJECTIVE METRICS.

**Aspects of Musicality and Acoustic Quality.**    To evaluate the subjective musical qualities of YuE and comparative models, we conducted a detailed A/B test on six dimensions: *vocal (acoustic) quality*, *accompaniment (acoustic) quality*, *music arrangement*, *melodic attractiveness*, *vocal-backtrack matching*, and *song structure*. We visualize the win rate with radar plot in Figure 14(L). Suno V4 consistently outperforms all other models across these aspects, thus we normalized the win rate by Suno to improve visual clarity. Among other models, YuE excels notably in music structure and music arrangement, highlighting its capability for coherent long-form composition capability. However, YuE shows clear deficiencies in vocal and accompaniment acoustic quality, likely due to limitations of its current audio tokenization method. While YuE achieves decent musicality and convergence, the semantic-fused tokenizer requires improvements in acoustic detail via an enhanced decoder or a super-resolution backend.

**Controllability.**    Similarly, we evaluated the controllability of YuE and comparative models through A/B testing on five dimensions: *genre control*, *instrument/vocal control*, *emotion control*, *tempo/rhythm control*, and *lyrics following*. Given limitations of existing classifiers and transcription systems, user preference win rate was our primary evaluation metric, with results presented in Figure 14(R). Suno v4 consistently outperforms all models across controllability metrics. Among other models, YuE performs strongest in genre adherence, instrument/vocal consistency, and emotion, highlighting its effectiveness in generating stylistically coherent music aligned with textual prompts. YuE demonstrates moderate performance in emotion and tempo control, indicating the need for improved lyric alignment and tempo tagging systems due to considerable noise observed in the pseudo label on the training corpus provided by Qwen2Audio. Overall, these results affirm YuE's robust controllability capabilities.

## H   CORRELATION BETWEEN AUTOMATIC METRICS AND HUMAN EVALUATION

**Correlation with Musicality & Average Preference**    When considering musicality and average human preference (Table 7), the **Vocal Range** metric stands out, correlating most strongly (above 0.85) with both subjective ratings. This highlights the crucial role of vocal expressiveness and melodic

Table 7: Pearson correlation between subjective metrics (*Musicality*, *Average*) and automatic metrics. **Vocal Range** strongly impacts Musicality and Average ratings.

|  | KL | FAD | CE | CU | PC | PQ | CLAP | CLaMP 3 | VocalRange |
|---|---|---|---|---|---|---|---|---|---|
| Musicality | -0.232 | -0.249 | 0.368 | 0.320 | -0.268 | 0.112 | -0.072 | 0.333 | **0.857** |
| Average | -0.199 | -0.351 | 0.357 | 0.303 | -0.128 | 0.054 | 0.086 | 0.264 | **0.858** |

diversity[21] in listeners' overall impressions of generated music. We find vocal range to be a practical proxy for musicality and recommend its adoption.

Table 8: Pearson correlation of alignment metrics vs. human preference on controllability.

|  | LyricFollow | GenCtrl | InstrCtrl | EmoCtrl | Tempo/RhyCtrl |
|---|---|---|---|---|---|
| CLAP↑ | -0.25 | 0.01 | -0.07 | 0.14 | 0.09 |
| CLaMP 3↑ | **0.42** | **0.37** | **0.44** | **0.33** | **0.36** |

**Alignment Metrics.** The correlation results in Table 8 demonstrate that CLaMP 3 scores consistently correlate better with human evaluations of controllability compared to CLAP scores. This is particularly evident in tasks such as LyricFollow (0.42 vs. -0.25) and InstrCtrl (0.44 vs. -0.07). Interestingly, the genre-following capability measured by the CLaMP 3 backend (Wu et al., 2025) appears to be closely related to lyric-following performance, even though lyrics are not explicitly included in the computation of the CLaMP 3 score. This indicates a correlation between genre controllability and lyric adherence in music generation models. Conversely, the weaker correlations observed with CLAP suggest limitations in its capacity to capture nuanced perceptual aspects, likely due to insufficient exposure to singing and music-specific content during pre-training.

Table 9: Pearson correlation of KL and FAD on acoustic quality preference metrics.

|  | AccompQual | VocalQual |
|---|---|---|
| KL | 0.14 | 0.23 |
| FAD | **-0.15** | **-0.11** |

**Distribution Matching Metrics.** We employed the more advanced PaSST (Koutini et al., 2021) backbone instead of the conventional VGGish (Hershey et al., 2017) to evaluate distribution matching metrics. Despite its sophistication, the AudioSet pre-trained backbone may inherently suffer from out-of-distribution (OOD) issues when dealing with generative music, particularly with singing or vocal elements. Additionally, sample size bias may contribute significantly, as limited availability of extensive audio samples from closed-source generative systems hinders accurate distribution estimations.

As shown in Table 9, both **KL** and **FAD** exhibit weak correlations with *accompaniment (acoustic) quality* (AccompQual) and *vocal (acoustic) quality* (VocalQual), suggesting that distribution-level metrics may not fully capture subtle subjective perceptions of acoustic fidelity in our case. However, as indicated in Table 7, these same metrics correlate more strongly with *musicality* and overall human preference[22]. This implies that while distribution matching may not always reflect finer acoustic details, they sometimes reflect qualities relevant to perceived musicality and listener satisfaction.

---

[21]One possible explanation relates to AR music generation behavior. Such models often favor high-probability tokens, biasing melodies toward conservative choices like tonic, chord tones, or previously generated notes. Poor optimization (e.g., overfitting) or overly conservative sampling exacerbates this issue, reducing melodic diversity.

[22]Both KL and FAD are negatively correlated, since lower values indicate better alignment.

**Content-Based Metrics.** In Table 10, **CE** exhibits the strongest correlations, particularly with subjective acoustic quality measures such as *VocalQual* (0.66) and *AccompQual* (0.56). This indicates that CE might be especially sensitive to acoustic characteristics perceived by listeners. By contrast, correlations with musicality-related aspects—such as *SongStruct* (0.33), *VAComp* (0.35), *MelAttrac* (0.30), and *MusicArr* (0.31)—are relatively lower, suggesting a lesser sensitivity of CE to detailed musical attributes. Meanwhile, both **PC** and **PQ** show notably weaker or inconsistent correlations across these subjective metrics, implying limitations in their ability to capture musicality related perceptual elements.

Table 10: Pearson correlation of content-based metrics vs. related preference metrics.

|     | AccompQual | VocalQual | SongStruct | VAComp | MelAttrac | MusicArr |
|-----|-----------|-----------|-----------|--------|-----------|----------|
| CE  | **0.56**  | **0.66**  | **0.33**  | **0.35** | **0.30** | **0.31** |
| CU  | 0.50      | 0.61      | 0.27      | 0.29   | 0.25      | 0.26     |
| PC  | -0.09     | 0.00      | -0.24     | -0.20  | 0.00      | -0.16    |
| PQ  | 0.27      | 0.36      | 0.05      | 0.06   | -0.03     | 0.02     |

## I    15 ENGLISH PROMPTS FROM GPT

---

**ID: 1**

[Genre] Rap

[verse]
Woke up in the morning, sun is shining bright
Chasing all my dreams, gotta get my mind right
City lights are fading, but my vision's clear
Got my team beside me, no room for fear
Walking through the streets, beats inside my head
Every step I take, closer to the bread
People passing by, they don't understand
Building up my future with my own two hands

[chorus]
This is my life, and I'm aiming for the top
Never gonna quit, no, I'm never gonna stop
Through the highs and lows, I'mma keep it real
Living out my dreams with this mic and a deal

[verse]
Late nights grinding, writing down these rhymes
Clock is ticking fast, can't afford to waste time
Haters gonna hate, but I brush it off
Turn the negativity into something strong
Mama working hard, wanna make her proud
Echoes of her prayers cutting through the crowd
Friends turned strangers, but it's all good
Focused on my path like I always knew I would

[chorus]
This is my journey, and I'm running this race
Heart full of fire, you can see it in my face
Obstacles ahead, but I got no fear
Victory is close, yeah, it's almost here

[bridge]
They said I couldn't do it, said I'd never rise
But now I'm soaring high, reaching for the skies
Lessons that I learned made me who I am

---

Standing tall now, I don't give a damn

[verse]
Echoes in the alley, music's getting loud
Feeling the adrenaline pumping through the crowd
Spotlights on me, it's my time to shine
Living in the moment, everything's aligned
Looking back now at the roads I've crossed
Every single battle, every line I've tossed
Made me stronger, wiser, ready for what's next
Writing my own story, turning pages of the text

[chorus]
This is my song, and I'm singing it proud
Voices united, hear us shout out loud
From the underground straight into the stars
Carving out my name, leaving all these scars

[outro]
Yeah, this is for the dreamers, the ones who never quit
Keep your head up high, and don't you ever submit
Life is what you make it, so make it something great
Step into your purpose, go and seize your fate

ID: 2

[Genre] Rock

[verse]
Standing on the corner, shadows in the night
The city's heartbeat echoes under lights
Hands deep in pockets, wandering alone
Footsteps tracing paths to the unknown
Suddenly he pauses, looks up to the sky
Eyes reflect the questions passing by
Whispers to the wind, words without a sound
Searching for the answers yet unfound

[chorus]
Lost within the chaos, seeking out a sign
In a world of color, drawing blurred lines
Moving forward, looking back, unsure of the way
Trying to find a place where he can stay

[verse]
He crosses empty streets, under neon glow
Faces in the crowd, stories left untold
Raises up his arms, reaching for the truth
Grasping at the fragments of his youth
Billboards and the banners flutter in the breeze
The rhythm of the city brings him to his knees
Heartbeat heavy, nowhere left to hide
Feeling like he's lost amidst the tide

[chorus]
Lost within the chaos, seeking out a sign
In a world of color, drawing blurred lines
Moving forward, looking back, unsure of the way
Trying to find a place where he can stay

[bridge]
Doesn't want to leave, doesn't want to fight
Caught between the darkness and the light
No need for reason, nothing to prove
Just a soul in transit, with nothing to lose

[outro]
Doesn't want to leave, doesn't want to fight
Chasing after shadows in the night
He doesn't need the truth, doesn't need a name
Just looking for a spark to fan the flame

ID: 3

[Genre] Pop

[verse]
Staring at the sunset, colors paint the sky
Thoughts of you keep swirling, can't deny
I know I let you down, I made mistakes
But I'm here to mend the heart I didn't break

[chorus]
Every road you take, I'll be one step behind
Every dream you chase, I'm reaching for the light
You can't fight this feeling now
I won't back down
I'm the whisper in the wind, the shadow by your side
The warmth you feel within when you can't hide
You know you can't deny it now
I won't back down

[verse]
They might say I'm foolish, chasing after you
But they don't feel this love the way we do
My heart beats only for you, can't you see?
I won't let you slip away from me

[chorus]
Every road you take, I'll be one step behind
Every dream you chase, I'm reaching for the light
You can't fight this feeling now
I won't back down
I'm the whisper in the wind, the shadow by your side
The warmth you feel within when you can't hide
You know you can't deny it now
I won't back down

[bridge]
No, I won't back down, won't turn around
Until you're back where you belong
I'll cross the oceans wide, stand by your side
Together we are strong

[outro]
Every road you take, I'll be one step behind
Every dream you chase, love's the tie that binds
You can't fight this feeling now
I won't back down

ID: 4

[Genre] Jazz

[verse]
In the quiet of the evening, shadows start to fall
Whispers of the night wind echo through the hall
Lost within the silence, I hear your gentle voice
Guiding me back homeward, making my heart rejoice

[chorus]
Don't let this moment fade, hold me close tonight
With you here beside me, everything's alright
Can't imagine life alone, don't want to let you go
Stay with me forever, let our love just flow

[verse]
Moonlight paints a picture upon your lovely face
Every glance between us fills the empty space
Time stands still around us when you're in my arms
Nothing else can matter, safe from any harm

[chorus]
Don't let this moment fade, hold me close tonight
With you here beside me, everything's alright
Can't imagine life alone, don't want to let you go
Stay with me forever, let our love just flow

[bridge]
Every touch ignites a fire, burning deep within
Every smile you give to me makes my head spin
Promise me you'll stay awhile, don't ever say goodbye
Together we'll chase every star across the sky

[chorus]
Don't let this moment fade, hold me close tonight
With you here beside me, everything's alright
Can't imagine life alone, don't want to let you go
Stay with me forever, let our love just flow

[outro]
Stay with me forever, let our love just flow

### ID: 5

[Genre] Blues

[verse]
Late last night, the rain was pouring down
Lonely footsteps echoed through the town
Thinking 'bout the love that slipped away
Wondering how I let you go that day

[chorus]
Oh, my angel, where have you flown
Left me here to face this world alone
I'm just a fool, a fool in love with you
Can't deny this heartache's true

[verse]
Streetlights flicker, shadows on the wall
Memories of you, I recall
Your laughter like a song inside my head
Without you here, my soul feels dead

[chorus]
Oh, my angel, won't you return
In this fire of love, I still burn
I'm just a fool, a fool in love with you
Hoping someday you'll feel it too

[bridge]
I fell for you, and I always knew
That my world revolves around you
I hope and I pray, both night and day
That you'll come back and choose to stay

[chorus]
Oh, my angel, where have you flown
Left me here to face this world alone
I'm just a fool, a fool in love with you
Waiting here, what else can I do

[outro]
I'm just a fool, a fool in love with you

## ID: 6

[Genre] RnB_Soul

[verse]
Why don't we just find a place to hide
Leave all our worries and doubts behind
When nothing in this world is as it seems
Together we can live inside our dreams
There's no need to be afraid tonight
In the love we've made, we'll find the light
When we're living in a world of our own
It's you and me, we never feel alone

[chorus]
They say it's hard for a man to let it show
But with you, I'm ready to let it all go
Whatever we try, we're gonna get there
You take control, baby, I don't care
I gotta keep on pushing when times get tough
We keep on making better love

[verse]
Don't believe the things that others say
We've tried it all and found our way
They should take a look at you and me
Learning how to love and set it free
For every heartache, we take our time
You teach me yours and I'll show you mine
About the way that love is meant to be
Together we'll rewrite our history

[chorus]
They say it's hard for a man to let it show
But with you, I'm ready to let it all go
Whatever we try, we're gonna get there
You take control, baby, I don't care
I gotta keep on pushing when times get tough
We keep on making better love

[bridge]
Gotta take control and swallow my pride
Every man has feelings deep inside
You gotta find yourself before you can
Be ready to love and understand
Baby, I know what you're thinking of
We keep on making better love

[outro]
I believe the love we're making's gonna last forevermore
Loving you feels so right, like never before
We'll be getting down tonight until the morning light
We keep on making better love
Better love, we'll be making
Better love, no more faking
They say it's hard for a man to let it show
But with you here, I'm ready to let go
Whatever we try, we're gonna get there
You take control, baby, I don't care
I gotta keep on pushing when times get tough
We keep on making better love
Better love (till fade out)

ID: 7

[Genre] Ancient_Chinese_Style

[verse]
Beneath the moonlit sky so vast
A lone wanderer recalls the past
Whispers of the bamboo leaves
Echo tales the wind retrieves

[chorus]
Oh, the rivers flow, mountains high
Journeying souls beneath the endless sky
Threads of fate entwine our way
Guiding us through night and day

[verse]
Lanterns glow with softest light
Painting shadows in the night
Silken robes and ancient songs
Memories where hearts belong

[chorus]
Oh, the rivers flow, mountains high
Journeying souls beneath the endless sky
Threads of fate entwine our way
Guiding us through night and day

[bridge]
Stars reflect in tranquil ponds
Dreams unfold of times beyond
Lotus blooms and cranes in flight
Secrets held within the night

[outro]
As the sunrise paints the east
Bringing hope and inner peace
Footprints fade upon the shore
But the spirit journeys evermore

ID: 8

[Genre] Folk

[verse]
Underneath the open sky so clear,
We gather 'round with voices near.
Through trials faced and stories told,
Our spirits rise, our hearts unfold.

[chorus]
So lift the lanterns to the sky,
Together we will soar and fly.
Though shadows loom and doubts appear,
We'll keep the flame forever here.

[verse]
Remember all the paths we've crossed,
The battles won, the moments lost.
A banner of hope we hold up high,
A symbol shining in our eyes.

[chorus]
So lift the lanterns to the sky,
Together we will soar and fly.
Though shadows loom and doubts appear,
We'll keep the flame forever here.

[bridge]
With hands united, we stand tall,
Pledged to rise if we should fall.
Through darkest nights and stormy seas,
Our song will carry on the breeze.

[chorus]
So lift the lanterns to the sky,
Together we will soar and fly.
Though shadows loom and doubts appear,
We'll keep the flame forever here.

[outro]
We'll keep the flame forever here.

ID: 9

[Genre] Dance

[verse]
Underneath the starlit sky,
We come alive, you and I.
City lights are shining bright,
Dancing through the endless night.

[chorus]
Who are we? Let's break away,
Feel the beat and let it play.
Lost in music, hearts align,
In this moment, we define.

[verse]
Shadows fade beneath the glow,
Rhythms guide us where to go.
Voices whisper in the crowd,
Turn it up, we'll sing aloud.

[chorus]
Who are we? Let's break away,
Feel the beat and let it play.
Lost in music, hearts align,
In this moment, we define.

[bridge]
Let the melody surround,
Lift us off the solid ground.
Every step and every move,
In this dance we find our groove.

[chorus]
Who are we? Let's break away,
Feel the beat and let it play.
Lost in music, hearts align,
In this moment, we define.

[outro]
Keep on dancing, feel the heat,
Moving to the pounding beat.
Who we are is here and now,
Take my hand, we'll show them how.

## ID: 10

[Genre] Country

[verse]
Da-dum, da-da-da-da-da-da-da
Da-dum, da-dum
Walking down this lonesome road
Thinking 'bout the love untold
Why haven't I told you
I've whispered to the midnight stars
Just how wonderful you are
Why haven't I told you

[chorus]
Friends keep asking if I'm fine
I just smile and say you're mine
Might as well confess
Can't keep this inside
Maybe you feel the same way too
Oh darling, if you do
Why haven't you told me
Da-dum, da-da-da-da-da-da-da

[verse]
I've sung it to the morning sun
That with you, my life's begun
Why haven't I told you
My heart's an open book today
Waiting for the words to say
Why haven't I told you

[chorus]
Friends keep asking what's the news
I just grin and think of you
Time to take a chance
Let my feelings show
Maybe you feel the same way too
Oh darling, if you do
Why haven't you told me

[bridge]
Da-dum, da-da-da-da-da-da-da
Da-dum, da-dum
No more holding back these words
Let them fly just like the birds

[chorus]
Now I'm standing here tonight
Hoping that I got it right
Might as well confess
Can't keep this inside
Maybe you feel the same way too
Oh darling, if you do
Let's not waste another day
Why haven't we told us

[outro]
Da-dum, da-da-da-da-da-da-da
Da-dum, da-dum
Now we've finally told us
Our new life's begun

ID: 11

[Genre] Rap

[verse]
Woke up in the morning, sun is shining bright
Chasing all my dreams, gotta get my mind right
City lights are fading, but my vision's clear
Got my team beside me, no room for fear
Walking through the streets, beats inside my head
Every step I take, closer to the bread
People passing by, they don't understand
Building up my future with my own two hands

[chorus]
This is my life, and I'm aiming for the top
Never gonna quit, no, I'm never gonna stop
Through the highs and lows, I'mma keep it real
Living out my dreams with this mic and a deal

[verse]
Late nights grinding, writing down these rhymes
Clock is ticking fast, can't afford to waste time
Haters gonna hate, but I brush it off
Turn the negativity into something strong
Mama working hard, wanna make her proud
Echoes of her prayers cutting through the crowd
Friends turned strangers, but it's all good
Focused on my path like I always knew I would

[chorus]
This is my journey, and I'm running this race
Heart full of fire, you can see it in my face
Obstacles ahead, but I got no fear
Victory is close, yeah, it's almost here

[bridge]
They said I couldn't do it, said I'd never rise
But now I'm soaring high, reaching for the skies
Lessons that I learned made me who I am
Standing tall now, I don't give a damn

[verse]
Echoes in the alley, music's getting loud
Feeling the adrenaline pumping through the crowd
Spotlights on me, it's my time to shine
Living in the moment, everything's aligned
Looking back now at the roads I've crossed
Every single battle, every line I've tossed
Made me stronger, wiser, ready for what's next
Writing my own story, turning pages of the text

[chorus]
This is my song, and I'm singing it proud
Voices united, hear us shout out loud
From the underground straight into the stars
Carving out my name, leaving all these scars

[outro]
Yeah, this is for the dreamers, the ones who never quit
Keep your head up high, and don't you ever submit
Life is what you make it, so make it something great
Step into your purpose, go and seize your fate

ID: 12

[Genre] Rock

[verse]
Standing on the corner, shadows in the night
The city's heartbeat echoes under lights
Hands deep in pockets, wandering alone
Footsteps tracing paths to the unknown
Suddenly he pauses, looks up to the sky
Eyes reflect the questions passing by
Whispers to the wind, words without a sound
Searching for the answers yet unfound

[chorus]
Lost within the chaos, seeking out a sign
In a world of color, drawing blurred lines
Moving forward, looking back, unsure of the way
Trying to find a place where he can stay

[verse]
He crosses empty streets, under neon glow
Faces in the crowd, stories left untold
Raises up his arms, reaching for the truth
Grasping at the fragments of his youth
Billboards and the banners flutter in the breeze
The rhythm of the city brings him to his knees
Heartbeat heavy, nowhere left to hide
Feeling like he's lost amidst the tide

[chorus]
Lost within the chaos, seeking out a sign
In a world of color, drawing blurred lines
Moving forward, looking back, unsure of the way
Trying to find a place where he can stay

[bridge]
Doesn't want to leave, doesn't want to fight
Caught between the darkness and the light
No need for reason, nothing to prove
Just a soul in transit, with nothing to lose

[outro]
Doesn't want to leave, doesn't want to fight
Chasing after shadows in the night
He doesn't need the truth, doesn't need a name
Just looking for a spark to fan the flame

**ID: 13**

[Genre] Pop

[verse]
Staring at the sunset, colors paint the sky
Thoughts of you keep swirling, can't deny
I know I let you down, I made mistakes
But I'm here to mend the heart I didn't break

[chorus]
Every road you take, I'll be one step behind
Every dream you chase, I'm reaching for the light
You can't fight this feeling now
I won't back down
I'm the whisper in the wind, the shadow by your side
The warmth you feel within when you can't hide
You know you can't deny it now
I won't back down

[verse]
They might say I'm foolish, chasing after you
But they don't feel this love the way we do
My heart beats only for you, can't you see?
I won't let you slip away from me

[chorus]
Every road you take, I'll be one step behind
Every dream you chase, I'm reaching for the light
You can't fight this feeling now
I won't back down
I'm the whisper in the wind, the shadow by your side
The warmth you feel within when you can't hide
You know you can't deny it now
I won't back down

[bridge]
No, I won't back down, won't turn around
Until you're back where you belong
I'll cross the oceans wide, stand by your side
Together we are strong

[outro]
Every road you take, I'll be one step behind
Every dream you chase, love's the tie that binds
You can't fight this feeling now
I won't back down

ID: 14

[Genre] Jazz

[verse]
In the quiet of the evening, shadows start to fall
Whispers of the night wind echo through the hall
Lost within the silence, I hear your gentle voice
Guiding me back homeward, making my heart rejoice

[chorus]
Don't let this moment fade, hold me close tonight
With you here beside me, everything's alright
Can't imagine life alone, don't want to let you go
Stay with me forever, let our love just flow

[verse]
Moonlight paints a picture upon your lovely face
Every glance between us fills the empty space
Time stands still around us when you're in my arms
Nothing else can matter, safe from any harm

[chorus]
Don't let this moment fade, hold me close tonight
With you here beside me, everything's alright
Can't imagine life alone, don't want to let you go
Stay with me forever, let our love just flow

[bridge]
Every touch ignites a fire, burning deep within
Every smile you give to me makes my head spin
Promise me you'll stay awhile, don't ever say goodbye
Together we'll chase every star across the sky

[chorus]
Don't let this moment fade, hold me close tonight
With you here beside me, everything's alright
Can't imagine life alone, don't want to let you go
Stay with me forever, let our love just flow

[outro]
Stay with me forever, let our love just flow

---

**ID: 15**

[Genre] Blues

[verse]
Late last night, the rain was pouring down
Lonely footsteps echoed through the town
Thinking 'bout the love that slipped away
Wondering how I let you go that day

[chorus]
Oh, my angel, where have you flown
Left me here to face this world alone
I'm just a fool, a fool in love with you
Can't deny this heartache's true

[verse]
Streetlights flicker, shadows on the wall
Memories of you, I recall
Your laughter like a song inside my head
Without you here, my soul feels dead

[chorus]
Oh, my angel, won't you return
In this fire of love, I still burn
I'm just a fool, a fool in love with you
Hoping someday you'll feel it too

[bridge]
I fell for you, and I always knew
That my world revolves around you
I hope and I pray, both night and day
That you'll come back and choose to stay

[chorus]
Oh, my angel, where have you flown
Left me here to face this world alone
I'm just a fool, a fool in love with you
Waiting here, what else can I do

[outro]
I'm just a fool, a fool in love with you

---

