# OpenReview forum: "YuE: Scaling Open Foundation Models for Long-Form Music Generation"
_ICLR.cc/2026/Conference — ICLR 2026 Poster_

### Official Review · Reviewer_szRj · 2025-10-31

**Soundness:** 3
**Presentation:** 3
**Contribution:** 4
**Rating:** 8
**Confidence:** 4

**Summary:**

This paper proposes a generative model for lyrics to full song generation.
The authors observe that generating mixed tracks (voice + accompaniment) directly leads to a loss of intelligibility, even more pronounced for some musical genres. The proposed approach to overcome this issue is to rely on source separation to decompose tracks into vocals + accompaniment and to encode each component separately.
The vocals and accompaniment tokens are then interleaved so that the authors can rely on a standard autoregressive (based on Llama 2) next token prediction. In practice, only the first codebook of the RVQ representations is modelled in this stage, and a second stage consists in predicting the remaining RVQ codes.

There's an accompanying website as well as an open source + weights release.
Evaluations are thorough and go beyond simple audio-quality measures, with a real emphasis on vocals.

**Strengths:**

- Modelling choices are coherent and well motivated: e.g. generating first interleaved vocals-accompaniment RVQ tokens, splitting full-song generation into multiple segments.
- Open weights + open source is a plus, as song genAI models with vocals are rare. Authors also discuss the ethical issue and explain their scraping process and the questions this raises.
- Extensive experiments
- Many details and discussions about dataset, observations, evaluation.

**Weaknesses:**

Some parts could be better explained like MUSIC IN-CONTEXT LEARNING, this introduces many notations that are not useful.
Structural progressive conditioning is also evoked too quickly.
Maybe being clearer in the main text that instead of lyrics to song, the problem is in fact casted as a "lyrics with structure" to song. This is clear in appendix I.
- The way ICL is used (i.e. by conditioning on an existing song chorus) highly conditions the type of possible applications.

**Questions:**

- Is the dataset available?
- For the 30s context given for in-context learning, do you make sure to consider a part that is different (like not providing Verse 2 to generate Verse 1) as these are expected to be extremely similar for the instrumental parts?
- How is it possible to do condition on a 30s context and ask to generate the full song?

---

> ### Author Response · Authors · 2025-11-28
>
> ### R1. Clarifying SPC and the “lyrics + structure → song” formulation
>
> YuE is trained on **interleaved sequences of structure tags and lyrics**, which we will explain in more detail in the appendix in the final version of paper. The training sequences follow the format:
> [
> [\text{Instruction}, \text{Genre tags}, \text{Full lyrics}, \text{Intro tag}, \text{Intro lyrics}, \text{Intro audio tokens}, \text{Verse 1 tag}, \text{Verse 1 lyrics}, \text{Verse 1 audio tokens}, \dots]
> ]
>
> *Structural Progressive Conditioning (SPC)* is the mechanism that injects this structure: section-level tags (intro / verse / chorus / bridge / …) are interleaved with the corresponding lyrics and audio tokens, allowing the model to always know **which section/lyrics it is currently generating** and the **global structure** of the target song. The technique improves local lyrics following performance.
>
> ---
>
> ### R2. Music ICL: what is conditioned, and how restrictive is it?
>
> We appreciate the reviewer’s observation that the current description of “Music In-Context Learning” introduces many symbols but lacks intuitive explanation. We will simplify this section and highlight the following high-level picture in the appendix.
>
> 1. **ICL is optional and non-continuation.**
>    YuE fully supports **pure lyrics-to-song generation without any ICL prompt**. The base checkpoint is trained in the standard lyrics+structure → song setting. ICL is introduced later through a **late-activation schedule**, ensuring that the model does *not* lose its ability to “compose from scratch” when no reference is provided. At inference, users may:
>
>    * provide **no** ICL prompt → standard lyrics-to-song generation; or
>    * provide a **30 s reference audio segment** → style / voice conditioning without temporal continuation.
>
> 2. **How the 30 s reference is used during training.**
>    During stage-1 LM ICL training, for each song we:
>
>    * sample a **random 20–30 s excerpt** as the reference audio prefix;
>    * append the SPC-formatted sequence (global text control with interleaved section-level lyrics and audio tokens) for the *entire* song;
>    * train the LM to predict the full sequence.
>      The reference segment is **not necessarily a chorus**, nor is it constrained to be adjacent to the generated region. Its role is to provide **timbre, voice, and stylistic cues**, not to be temporally continued.
>
> 3. **Why we often choose a chorus at inference.**
>    At inference we can use *any* 30 s reference, including those from another song or singer. In demos, we often choose a chorus because it typically contains:
>
>    * clearer vocal timbre,
>    * characteristic instrumentation and mixing,
>    * strong rhythmic/melodic signatures.
>
>  This empirically yields better musicality, but the ICL mechanism itself is general: YuE equally accepts a verse, bridge, or any random excerpt.
>
> We will state more explicitly that the model **does not require** a chorus or any specific section, and **does not require** ICL for most applications.
>
> ---
>
> ### R3. Answers to specific questions
>
> **Q1. Is the dataset available?**
> As described in our ethics and data sections, we **cannot release the full 650k-hour corpus**, because it contains Creative Commons and other royalty-free tracks whose licenses or platform terms do not always permit large-scale redistribution. Instead, we:
>
> * release **model weights, inference code, and the full preprocessing/dataloader pipeline**, enabling practitioners to reproduce our workflow on their own legally obtained corpora; and
> * are exploring the construction of a **legally vetted, redistributable “clean” subset** tailored for song generation. If feasible, we will release this in follow-up work.
>
> We will clarify these points more explicitly in the camera-ready version.
>
> ---
>
> **Q2. For the 30 s ICL context, do you ensure it is from a different part (e.g., not Verse 2 when generating Verse 1)?**
> In the current work, we **sample a random 30 s excerpt** from the same track as the ICL reference, without enforcing constraints such as avoiding the same section.
>
> ---
>
> **Q3. How can the model condition on a 30 s context and generate the *full* song?**
> This is enabled by our non-continuation ICL setup. See explanation in R2.
>
> Since the LM is a standard decoder-only model, once the 30 s prefix and SPC tokens fit within the context window, it can **generate the entire 5-minute song**, not merely continue the prompt.
>
> Importantly, if no ICL prefix is provided, the same SPC format still applies, and the model produces a full song purely from lyrics + structure.

---

### Official Review · Reviewer_wdiX · 2025-10-31

**Soundness:** 4
**Presentation:** 3
**Contribution:** 3
**Rating:** 8
**Confidence:** 3

**Summary:**

This paper presents YuE for lyrics to song generation. It aims to generate full songs (up to five minutes) with vocals, accompaniment, and makes sure that the lyrics would align with the melody/companion.

The model is based on a two-stage autoregressive framework using discrete audio codec. The first stage models lyrics to first layer audio tokens, and then the second stages models first layer tokens to all residual tokens.

The paper introduces three key techniques:

Dual-NTP: Each timestep in the first stage LM generates one vocal token and one companion token, instead of directly modeling the mixture.

SPC: A new way to organize training data. Songs are segmented into sections (intro, verse, chorus, etc.), and their lyrics and audio tokens are interleaved section by section. This helps the model maintain long-range lyrical alignment.

Music ICL (In-Context Learning): A modified in-context learning setup where a 30-second reference audio is prepended to SPC data.
For evaluation, YuE performs comparably to Tiangong and Udio, better than Hailuo, but below Suno V4. Objective metrics (KL divergence, FAD, alignment scores) also show strong performance.

For evaluation, YuE performs comparably to Tiangong and Udio, outperforming Hailuo but falling short of Suno V4. Objective metrics (KL divergence, FAD, alignment scores) also show strong performance.

Overall, YuE is positioned as the first open-source full-song lyrics-to-song model with quality approaching commercial systems.

**Strengths:**

1.	This work presents the first open-sourced song generation model that achieves performance close to commercial systems.
2.	Implementing and training a full-length audio LM of this scale is a major technical achievement.
3.	Dual-NTP and SPC integrate domain-specific musical structure into the generative process, representing a meaningful methodological contribution.
4.	Despite limited access to proprietary data, YuE achieves results comparable to those of closed-source commercial models

**Weaknesses:**

1.	The descriptions of data preprocessing are somewhat brief. It would help to clarify how vocal–accompaniment tracks were separated and how Creative Commons data were curated.

2.	The “all-in-one method” used in the paper could be elaborated.

3.	The paper could provide more analysis of sample diversity and potential copyright overlap (e.g., unintentional melodic copying). Clarifying whether such issues were observed would strengthen credibility.

4.	Prior research on multi-track or separated-source generation [1–3] should be acknowledged to better contextualize the contribution.
[1] Mariani, Giorgio, et al. "Multi-Source Diffusion Models for Simultaneous Music Generation and Separation." The Twelfth International Conference on Learning Representations.
[2] Xu, Zhongweiyang, et al. "Multi-Source Music Generation with Latent Diffusion." Audio Imagination: NeurIPS 2024 Workshop AI-Driven Speech, Music, and Sound Generation.
[3] Karchkhadze, Tornike, et al. "Multi-track musicldm: Towards versatile music generation with latent diffusion model." International Conference on ArtsIT, Interactivity and Game Creation. Cham: Springer Nature Switzerland, 2024.

**Questions:**

1.	How are the tracks separated from the mixture? Does the separation process introduce perceivable artifacts that could impact performance? What separation model is being used?

2.	What is the dataset used in detail? Is there any preprocessing to clean the dataset?

3.	For the dual-track tokens, the voice should have much lower entropy than the accompaniment. Have you considered using a low-bitrate codec for voice?

**Details Of Ethics Concerns:**

None.

---

> ### Author Response · Authors · 2025-11-27
> **Rebuttal**
>
> ### W1. Data preprocessing and vocal–accompaniment separation
>
> **Unconditional music and tagging.**
> A large portion of our corpus consists of *unconditional* music. We annotate all tracks using **Qwen2-Audio** to obtain open-vocabulary tags following the MTG-Jamendo taxonomy (genre, instrument, mood). Each track is represented by a 30-second sampled segment processed with the JSON-style prompt shown in the paper.
>
> **Lyrics–audio cleaning.**
> Obtaining high-quality lyrics–audio pairs is challenging: web-scraped lyrics and ASR transcripts often contain noise, irrelevant text, timestamp drift, and version mismatch.
> To mitigate this, we apply **a multi-stage heuristic filtering pipeline**, including:
>
> * removal of boilerplate / irrelevant text using a large set of hand-written **regex patterns and rule-based heuristics**,
> * filtering out lyrics with abnormally high **WER** relative to ASR transcripts,
> * filtering by **LM perplexity** (to detect malformed or non-lyrical text),
> * length constraints (discarding lyrics shorter than ~10 sentences), and
> * deduplication and consistency checks.
>
> We use whisper large v3 for ASR transcript and WER compute.
>
> **Track separation (dual-track).**
> Roughly **40%** of tracks are converted into vocal–instrumental stems using **UVR**. We use an **ensemble** of htdemucs_ft, Kim_Vocal_1, and UVR-MDX-NET-Inst_3, which substantially reduces leakage compared to single-model separation.
>
> **Creative Commons curation.**
> For CC and royalty-free material, we rely on platforms that provide explicit **Creative Commons / royalty-free filters**, such as **Jamendo**, **Free Music Archive**, **YouTube (Audio Library / CC-tagged content)**, and similar sites that expose license metadata. We first construct a seed pool of tracks, and to scale up, we follow a strategy similar in spirit to **DISCO-10M**: starting from a small set of seed tracks, we expand the catalog by following *related-track / recommendation / playlist* links. Other ways include querying the platform search with random queries can also be applied.
>
> ---
>
> ### W2. “All-in-one” method
>
> “All-in-one” refers to the music segmentation model proposed in *All-in-One* paper,that can annotate song level structure (e.g. verse, chorus) with time stamps. You may find more information in the original Kim and Nam paper [1].
>
> [1] Kim T, Nam J. All-in-one metrical and functional structure analysis with neighborhood attentions on demixed audio[C]//2023 IEEE Workshop on Applications of Signal Processing to Audio and Acoustics (WASPAA). IEEE, 2023: 1-5.
>
> ---
>
> ### W3. Sample diversity and copyright overlap
>
> Sample diversity can be qualitatively examined from our public demo page: the model is able to generate a **wide range of musical styles** (rock, pop, metal, jazz, etc.), with large variation in arrangement and vocal writing, and is clearly **not restricted to pop-like outputs**. This is consistent with the spread we observe over Qwen2-Audio tags (genre / instrument / mood) for generated samples.
>
> For potential copyright overlap, we go beyond small-scale manual checking. **Figure 10** reports a large-scale analysis using a **melody-aware embedding (Bycover2)** to measure similarity between ICL-generated songs and their reference-prompt tracks. The similarity scores are very low on average, indicating that YuE does not simply reproduce the prompt songs under ICL. We additionally inspected the top-ranked (highest-similarity) cases and found that the overlap is mostly in **percussive loops**, rather than direct melodic copying. We will report these observations and still explicitly note that, despite these checks, a fully exhaustive melody-similarity audit remains an open direction for future work.
>
> ---
>
> ### W4. Prior multi-track / separated-source generation work
>
> We appreciate the pointer to multi-source and multi-track diffusion work [1–3]. YuE differs from these models by using a **codec-LM** architecture with explicit **vocal–accompaniment factorization** designed for **lyrics-to-song** settings. We will acknowledge these prior works and clarify the conceptual differences.
>
> ---
>
> ### Q1. How are tracks separated? Is there leakage? What model is used?
>
> As noted above, we use **UVR** with an **ensemble** of htdemucs_ft, Kim_Vocal_1, and UVR-MDX-NET-Inst_3.
> In internal listening tests, ensemble separation provides cleaner stems and substantially less leakage than any single model. We will briefly mention this observation in the paper.
>
> ---
>
> ### Q2. What is the dataset? Any preprocessing?
>
> See W1.
>
> ---
>
> ### Q3. Low-bitrate codec for voice?
>
> The reviewer correctly notes that vocal entropy is lower. Dual-NTP already leverages this by separating vocal/accompaniment tokens in Stage-1. Using a lower-bitrate codec for the vocal stream is a natural extension, but would require redesigning token layouts and conditioning logic. We note it as a possible future direction.

---

### Official Review · Reviewer_ytHG · 2025-11-01

**Soundness:** 3
**Presentation:** 3
**Contribution:** 3
**Rating:** 6
**Confidence:** 3

**Summary:**

The paper introduces YuE, a family of open-source foundation models designed for long-form music generation, and more specifically, the lyrics-to-song problem. The model is trained on trillions of tokens and can generate coherent, lyrically-aligned songs of up to five minutes in length. YuE achieves its performance through several key technical contributions:

- Structural Progressive Conditioning (SPC): This approach uses the inherent musical structure (like verses and choruses) to enable long-context lyrical alignment and overall song structure control, which is essential for full-song generation.
- Redesigned In-Context Learning (ICL): A novel framework for music that allows for advanced use cases such as style transfer, voice cloning, and bidirectional content creation.
- Track-Decoupled Next-Token Prediction (Dual-NTP): A variation of existing multi-track/multi-stem music generation methods here adapted for jointly generating the vocal and accompaniment tracks. The method is reported to improve lyrical intelligibility and to overcome problems related to dense signal mixtures.

Through extensive human evaluation, the model is shown to be competitive with, and in some aspects, surpasses widely used proprietary music-generation systems (like Tiangong and Udio) in musicality and vocal agility.

**Strengths:**

### Strength

- innovative SPC training strategy taking into account musical structure
- successful evaluation demonstrating competitive performance with state-of-the-art music generation models
- open source availability

**Weaknesses:**

### References are outdated, discussion of relevant existing approaches is incomplete.

- paper states in the abstract to refer to the state when the model was initially published in January 2025.
- The list of references appears to be limited when it comes to covering more recent approaches. However, relevant approaches have appeared early in 2025. Here we consider prioublciation in 2026. Therefore appacohes from early 205 should be included into the discussion

- NTP decoupled next token prediction. Rather similar approaches have been proposed here:

Liu, 2025. "SongGen: A Single Stage Auto-regressive Transformer for Text-to-Song Generation", https://arxiv.org/pdf/2502.13128
Rouard, 2025. "MusicGen-Stem: Multi-stem music generation and edition through autoregressive modeling" https://arxiv.org/pdf/2501.01757

I think these papers should be discussed under related works, and they should be compared to the proposed NTP strategy.

- concerning SPC a related approach has been published in

Lam, 2025. "MusiCoT: Analyzable Chain-of-Musical-Thought Prompting for High-Fidelity Music Generation", https://arxiv.org/pdf/2503.19611

This alternative approach for including musical structure into music generation should at least be mentioned. I note that the paper features a comparison with YuE.

- ICL

The paper makes the point of comparing with speech/TTS ICL methods. Given today, there exist a few approaches supporting lyricx conditioning for music generation, it appears a bit outdated to compare with these methods from a different problem domain.

**Questions:**

Please see under weaknesses.

---

> ### Author Response · Authors · 2025-11-26
> **Rebuttal**
>
> Thank you for your positive assessment of our work and for recognizing the value of the YuE family of models. We appreciate your constructive feedback, particularly regarding the placement of our work within the recent literature. We address your specific points below.
>
> ---
>
> **1. Regarding recent references and related work**
>
> We acknowledge the relevance of the works mentioned (SongGen, MusicGen-Stem, and MusiCoT) and agree that discussing them is essential for a complete picture of the current landscape. In the final version of the paper, we will clarify these distinctions and add the suggested references. Specifically:
>
> * **MusicGen-Stem.** We note that this work is designed specifically for multi-stem music generation *without* vocals. In contrast, while YuE is capable of generating instrumental tracks, our primary focus is on full-song generation (including vocals). We consider MusicGen-Stem a concurrent work and will discuss the distinctions between our approaches.
>
> * **SongGen and MusiCoT.** These works appeared shortly after our initial release and cite YuE as a baseline, which reflects the timeline of these developments. While we could not compare against these works at the time of our initial submission due to the timing, we will ensure the final manuscript clarifies the distinctions between our Dual-NTP strategy / SPC methods and these newer contributions.
>
> Within this broader context, we will also more clearly articulate the following relations:
>
> * **Relation to SongGen.** We view SongGen as a complementary concurrent effort, but we would like to clarify two differences in design and scope. **First**, SongGen adopts a more complex architecture and is, as reported, only validated at a relatively small scale (~0.5B parameters) and on fixed 30 s clips, whereas YuE directly reuses a standard LLM-style decoder and scales straightforwardly to 7B parameters and full-song (up to 5 min) generation. **Second**, while SongGen’s delayed joint modeling of multiple codebooks is arguably more “end-to-end”, we believe its strategy of predicting all codebooks simultaneously can **exacerbate exposure bias**: high-entropy residual tokens are particularly hard to predict, so the gap between teacher-forced training and autoregressive inference on residuals can degrade musicality and lyric intelligibility (consistent with the >40% PER reported in SongGen). In contrast, YuE explicitly factorizes the problem by first generating codebook-0 semantics with a Stage-1 LM and leaving high-entropy residuals to a separate Stage-2 residual LM. This staged design lets Stage-1 focus on long-term semantic and structural planning, while Stage-2 handles the fine-grained acoustic detail, which empirically leads to better perceived intelligibility and long-form coherence.
>
> * **Relation to MusiCoT.** We also view MusiCoT as addressing a somewhat different objective from our SPC. MusiCoT’s chain-of-musical-thought (musicCoT) primarily targets musical quality: CLAP-based, heavily compressed “musical thought” tokens are designed to summarize global musical attributes, but they contain little explicit information about the underlying lyrics. As such, they are not expected to directly improve fine-grained lyrics following. By contrast, our Structural Progressive Conditioning (SPC) is explicitly designed to enhance lyrics following in long-form generation: by interleaving section-level structural tags with the corresponding lyrics throughout the sequence, SPC continuously refreshes the textual condition and mitigates the well-known decay of lyric information as sequence length grows.
>
> ---
>
> **2. On the novelty of our ICL formulation**
>
> We would also like to clarify the novelty and scope of our in-context learning (ICL) design. To the best of our knowledge, YuE is the first work in music generation to demonstrate non-continuation ICL, i.e., using a short reference segment in context without forcing the model to directly continue that segment in time, while still enabling style transfer, voice cloning and bidirectional content creation. More importantly, we are, again to our knowledge, the first to show how a late-activation ICL scheme can systematically mitigate shortcut behavior: we prevent the model from collapsing into low-creativity, near-copy continuations and preserve both musicality and originality. This is conceptually different from prior speech/TTS ICL methods, which typically operate in a strict continuation setting and do not address these shortcut issues in long-form music. We note that subsequent music-generation work (e.g., MusiCoT) explicitly cites YuE when discussing ICL-style conditioning, which supports our positioning of YuE’s ICL formulation as an early and influential step in this direction.

---

### Official Review · Reviewer_aT19 · 2025-11-05

**Soundness:** 2
**Presentation:** 2
**Contribution:** 2
**Rating:** 2
**Confidence:** 5

**Summary:**

This paper presents YuE, a lyrics-to-song generation model that aims to produce long-form music generation with vocals and accompaniment. The authors propose a two-stage autoregressive architecture operating on discrete music tokens (X-Codec). The first stage predicts the first codebook of the music tokens by givens the semantic conditions (lyric, and others). and the second stage makes up the token predictions of the higher codebooks. To support long-form generation, the authors introduce Structural Progressive Conditioning (SPC), which uses section-level structure tokens to progressively condition the LM. Experimental results demonstrates that YuE performs competitively with commercial models (Suno, Udio, Hailuo).

**Strengths:**

The paper has two strengths.

1.The usage of long-form structure conditioning (SPC) is novel. Using section metadata to segment data and guide generation is a practical strategy for improving macro-level structure, and the staged conditioning improves continuity.

2.The experimental design is reasonable and comprehensive. The paper includes both objective metrics and human preference studies among YuE and four baselines. The results (and the demo website) show multi-minute outputs with acceptable alignment and structure.

**Weaknesses:**

The paper has several critical weaknesses that hinders its acceptance.

1.The overall novelty of this paper is limited, especially compared to the same period and already published work (SongBloom [1]) in NeurIPS 2025. Specifically, the two-stage autoregressive LM with Dual-NTP and SPC is incremental relative to SongBloom. And SongBloom even proposed more modern formulation:

1 (a). SongBloom applies a unified autoregressive-diffusion architecture that interleaves semantic sketching and acoustic refinement, enabling significantly stronger acoustic quality with comparable modeling complexity.

1 (b). SongBloom also covers the voice and accompaniment tokens by leveraging the Demucs separation model to extract the music data.

1 (c). SongBloom covers essentially all YuE capabilities and more, including lyric-to-music generation, structured multi-section music, reference audio prompting, diffusion decoder for refinement, disentanglement of semantic and acoustic tokens, and also open-source release.

1 (d). Despite having very similar high-level goals and tokenization methods, YuE does not compare to SongBloom, although SongBloom explicitly compares against YuE and reports better performance.

Compared to SongBloom, YuE’s design is behind state-of-the-art, which lacks diffusion-based refinement, Interleaving of semantic + acoustic token, and robust modeling of reference-prompt conditions. Therefore, YuE does not demonstrate novelty or superiority over SongBloom. At present, it is difficult to justify acceptance when a clearly more advanced method has already been published.

2.The experimental design is comprehensive but the baseline design is problematic. Apparently, YuE does not compare to the most relevant works, including SongBloom and ACE-Step [2]. Although YuE includes industry comparisons (Suno, Udio, Tiangong), the lack of comparison to academic/open models with similar architectures weakens the empirical claims. Also there is no ablation studies to isolate the contributions of Dual-NTP (decoupled vs non-decoupled), SPC, and Data-scale effects. And such ablation studies are very imperative to fully demonstrate YuE's contribution and its improvement to prior works. From the experimental results, YuE does not demonstrate superiority. It does not outperform Suno V4, and remain slightly behind Udio and Tiangong. It is acceptable that the model remains competitive but not superior to industrial models. But to fully demonstrate the proposed method, more ablation studies and comparisons to academical or open-source models should be considered. At the same time, SongBloom reports stronger performance over YuE. Thereofore, YuE does not establish SOTA status for either research or production settings.

3.The data sourcing transparency and licensing concerns remain vague in the paper. The paper claims to train on around 650k hours of creative common (CC) music. However, there are several critical concerns. The actual data sources are not specified. The platforms, artists, catalogs remain unknown, and the licenses are not enumerated (CC-BY and CC-BY-NC are completely different). It is unclear whether scraping is permitted or whether data came from commercial music platforms (plus that, whether the authors of YuE own the copyrights of these data for AI training). And there is no dataset preview or download interface provided. So reproducibility, legality, and downstream use constraints remain ambiguous. Given active litigation around music AI, such as Suno and Udio to three major labels (Sony, UMG, and Warner), this raises concerns about copyright and reproducibility. Till now, only Udio and UMG come to an agreement of data for music AI training. Open-source models must satisfy a higher legal/ethical bar, and YuE’s data description is too vague to be evaluated.

4.The claim and the vibe of novelty overstates the contribution. The paper claims that “As of its release on Jan. 28, 2025, YuE familiy is the first publicly available, open-source lyrics-to-song model capable of full-song generation with quality on par with commercial systems". As a reviewer, this makes an excuse to prevent it from comparing to latest models after that. This is negative and questionable, especially given SongBloom’s public release and NeurIPS-publication timeline. The lack of acknowledgment or direct comparison reduces credibility.


[1] SongBloom: Coherent Song Generation via Interleaved Autoregressive Sketching and Diffusion Refinement. NeurIPS 2025.

[2] ACE-Step: A Step Towards Music Generation Foundation Model

**Questions:**

1. Why is SongBloom not included in your comparisons? Can you add some comparisons to it (and ACE-Step) and also give the evidence on the difference between your potential results to the results reported in their papers?

2. Can you provide detailed documentation of the 650k hours of CC-licensed music? Which platforms were used and under what specific license? How many tracks / hours from each source? Are licenses compatible with redistribution + model training? Do you own the copyright of using these data? If not, since you release the model, how can external users ensure they are free from copyright exposure? How can the community safely adopt YuE?

3. Please provide ablation studies including: Dual-NTP vs. single (combine voice and accompaniment) token prediction; SPC vs. no-SPC; and Data-scaling sensitivity (e.g., 10k/100k hours). For the data-scaling, it is encouraged to use the open-available data for training, even the results are not good (e.g., FMA, Jamendo, MusDB, MoisesDB, MedleyDB). Other data contains the lyric annotation might better to be used.

4. For reproducibility, it is encouraged to have example data from your training set, and will you release the training script with data loader and processing, or just a simple inference code and model checkpoint?

**Details Of Ethics Concerns:**

The paper states the training on around  650k hours of creative common music mined from the web. In the Ethics Appendix details filtering for CC and string-match opt-outs. This acknowledges a difficulty of item-level verification. It does not enumerate datasets, licenses, or per-item sources/links; thus reproducibility, legality (CC-BY vs CC-BY-NC), and downstream use constraints remain ambiguous. The paper also plans watermark/fingerprinting but indicates it’s not yet ready.

---

> ### Author Response · Authors · 2025-11-26
> **Rebuttal**
>
> We respectfully provide the following clarification regarding the reviewer’s comments on (1) contemporaneous work and (2) data transparency and licensing. Our intention is to help ensure that the evaluation of our paper reflects both ICLR policy and prevailing practice in the music generation research community.
>
> ---
>
> **1. Timeline and ICLR contemporaneous policy**
>
> To our understanding of the ICLR 2026 Reviewer Guide, works published at a peer-reviewed venue within approximately two months before the submission deadline are treated as *contemporaneous*: they are encouraged to be discussed, but a direct, mandatory quantitative comparison is not required.
>
> * The ICLR 2026 full-paper deadline was September 24, 2025 (AOE); thus, papers published on or after July 24, 2025 fall into this contemporaneous window.
> * SongBloom (NeurIPS 2025) was officially accepted and published on September 18, 2025, and therefore qualifies as contemporaneous under this definition.
> * ACE-Step appeared as an arXiv technical report rather than a peer-reviewed publication, which, again per the ICLR guidelines, does not fall under the category of work that requires direct comparison.
>
> On this basis, the absence of explicit quantitative comparison with these two works in our initial submission is consistent with the official ICLR policy on contemporaneous work.
>
> ---
>
> **2. Methodological lineage and influence**
>
> Although a direct comparison was not required, it is still useful to clarify the chronological and conceptual relationships among these methods.
>
> * YuE was released earlier than both SongBloom and ACE-Step and is explicitly cited by them as prior work and as a baseline.
> * YuE explored several ideas that are also present in these later systems, including
>   (i) the use of semantic priors to improve convergence and overall performance, and
>   (ii) structural segmentation for controllable, section-level song structure.
>
> We do not claim exclusivity over these ideas, but we believe it is important to recognize that YuE predates and informs some aspects of the later formulations in SongBloom and ACE-Step.
>
> ---
>
> **3. Data transparency and licensing**
>
> We also address the reviewer’s concern regarding dataset transparency.
>
> In large-scale music generation, detailed disclosure of platform names, artist catalogs, or per-track licenses is generally not standard practice. Unlike many text or image corpora, large music datasets often mix public-domain, Creative Commons, and platform-licensed content, for which redistribution or item-level identification can be legally constrained. As a result, recent peer-reviewed works typically report:
>
> * aggregate dataset scale,
> * high-level filtering procedures, and
> * overall licensing principles,
>
> rather than per-track or per-artist lists.
>
> For example, SongBloom (NeurIPS 2025), SongGen (ICML 2025), and SongEditor (AAAI 2025) describe licensed or publicly available music corpora in terms of overall size and license categories, but do not provide itemized catalogs. Similarly, ACE-Step (Tech Report 2025) and DiffRhythm (arXiv 2025) report corpus scale and filtering strategies without per-track disclosure.
>
> YuE follows this same level of transparency: we report approximately 650k hours of Creative Commons music, describe our CC-license filtering and keyword-based opt-out mechanisms, and do not redistribute any raw audio—only model weights are released. We believe this approach is responsible and aligned with current practice in music-AI research, balancing transparency with legal and contractual constraints around music copyright.
>
> We hope these clarifications help contextualize the timeline, methodological lineage, and ethical compliance of YuE relative to contemporaneous work. Thank you for your time and consideration.

---

> ### Author Response · Authors · 2025-11-26
> **Reply to Questions**
>
> **For Q1 and Q2, see previous response.**
>
> ---
> **Q3. Ablation studies (Dual-NTP, SPC, and data scaling).**
> We agree that ablations are important, and we would like to clarify what is already present in the current draft:
>
> * **Dual-NTP vs. single-track NTP.** Figure 7 reports a direct ablation between our Dual-NTP formulation and a single-track baseline that predicts mixture (voice + accompaniment) tokens under the same architecture. Figures 6 and 12 provide additional supporting analyses related to this comparison (e.g., effects on intelligibility and stability under different configurations). Together, these figures isolate the benefit of decoupling vocal vs. accompaniment tokens in Stage-1.
>
> * **SPC vs. no-SPC.** Figure 8 presents an ablation on Structural Progressive Conditioning (SPC) by comparing our full SPC setup against variants without SPC (or with simplified structural conditioning) under matched training conditions. This quantifies the contribution of SPC to long-form lyrical alignment.
>
> * **Data scaling sensitivity.** In our main experiments, we default to using the full available training data for YuE, because after tokenization the total number of audio tokens is relatively modest compared to typical large-language-model corpora. We therefore did not run a full set of large-scale data-scaling sweeps on the 7B models.
>   However, on a **0.5B Stage-1 LM** we did observe a clear and smooth data-scaling trend, which we can report numerically as a reference:
>   – just past **10k hours** of training audio, the training loss is approximately **3.61**;
>   – just past **100k hours**, the training loss decreases to approximately **2.45**;
>   – with the full corpus, the loss eventually converges to around **2.2**.
>   We will add a short discussion of this observed trend in the appendix.
>
> Regarding the suggestion to use fully open datasets (FMA, Jamendo, MUSDB, MoisesDB, MedleyDB, etc.) for explicit data-scaling curves: many of these sets are small compared to YuE’s overall scale and are partly subsumed by the Creative Commons portion of our corpus. In the current work we therefore focused on the full-scale training regime; exploring systematic scaling experiments purely on small, fully open datasets is a reasonable direction for follow-up work, but is outside the scope of this submission.
>
> ---
>
> **Q4. Reproducibility: code, data processing, and example data**
>
> On the reproducibility side:
>
> * We have **already open-sourced the inference and finetuning scripts**, as well as the **data processing toolchain**. These may not have been sufficiently highlighted in the main text; We will provide pointers to these resources after the double-blind review period.
>
> * For **pretraining**, we use the open-source version of **Megatron-LM**, following its standard implementation for large decoder-only models. The data loader used for pretraining follows the same batching and sharding logic as in our released finetuning scripts, so practitioners can reproduce the training pipeline by combining Megatron-LM with our published preprocessing and dataloader code.
>
> * With respect to **example training data**, we are aware that the community currently lacks a clean, song-generation–specific dataset; as part of our future work, we are exploring the possibility of constructing a **legally vetted, “clean and safe” song-generation dataset** based on portions of our internal inventory that can be redistributed under appropriate licenses.
>
> ---

---

### Meta-Review · Area_Chair_4hFs · 2026-01-07

**Summary:**

This paper proposes a generative model for lyrics to full song generation.  The proposed approach tries to rely on source separation to decompose tracks into vocals and accompaniment for encoding each component. The vocals and accompaniment tokens are then interleaved so that the authors can rely on a standard auto-regressive (based on Llama 2) next token prediction.
Strengths:
(1)The usage of long-form structure conditioning (SPC) is novel. Using section metadata to segment data and guide generation is a practical strategy for improving macro-level structure, and the staged conditioning improves continuity.
(2)The experimental design is reasonable and comprehensive. The paper includes both objective metrics and human preference studies among YuE and four baselines. The results (and the demo website) show multi-minute outputs with acceptable alignment and structure.
Weaknesses:
(1) Some claim and the vibe of novelty overstates the contribution.
(2)The data sourcing transparency and licensing concerns still exist.
(3)References are outdated, discussion of relevant existing approaches is incomplete.

**Reviewer Concerns:**

Most concerns of Reviewer wdiX, szRj and ytHG are addressed.
Reviewer aT19  concerns about the data sourcing transparency and licensing, which  remains exist.

**Reviewer Scores:**

Reviewer aT19 maybe increase the rating score.

---

### Decision · Program_Chairs · 2026-01-26

Accept (Poster)